# Cryo-EM structures of thermostabilized prestin provide mechanistic insights underlying outer hair cell electromotility

Haon Futamata[1], Masahiro Fukuda [1,5], Rie Umeda[1], Keitaro Yamashita [1,6], Atsuhiro Tomita[1], Satoe Takahashi [2], Takafumi Shikakura [3], Shigehiko Hayashi[3], Tsukasa Kusakizako [1], Tomohiro Nishizawa [1,7] ✉, Kazuaki Homma [2,4] ✉ & Osamu Nureki [1] ✉

Outer hair cell elecromotility, driven by prestin, is essential for mammalian cochlear amplification. Here, we report the cryo-EM structures of thermo-stabilized prestin (Pres[TS]), complexed with chloride, sulfate, or salicylate at 3.52-3.63 Å resolutions. The central positively-charged cavity allows flexible binding of various anion species, which likely accounts for the known distinct modulations of nonlinear capacitance (NLC) by different anions. Comparisons of these Pres[TS] structures with recent prestin structures suggest rigid-body movement between the core and gate domains, and provide mechanistic insights into prestin inhibition by salicylate. Mutations at the dimeric interface severely diminished NLC, suggesting that stabilization of the gate domain facilitates core domain movement, thereby contributing to the expression of NLC. These findings advance our understanding of the molecular mechanism underlying mammalian cochlear amplification.

Hearing is a process in which sound-induced mechanical vibrations are converted into electric signals within the cochlea and relayed to the brain. Two types of auditory hair cells, the inner (IHCs) and outer hair cells (OHCs), reside in the organ of Corti in the cochlea. The stereocilia of both hair cell types are mechanosensitive and transduce sound-induced mechanical displacement of the organ of Corti into changes in the receptor electric potential. IHCs are the direct sound sensory receptors that convert receptor potential changes into glutamatergic stimulation of the auditory nerves, whereas OHCs convert the receptor potential changes into somatic motility, which is referred to as OHC electromotility[1]. OHC electromotility mechanically amplifies the sound-induced displacement of the organ of Corti, which is otherwise attenuated by the viscous damping of the cochlear fluid. OHC electromotility plays an essential role in mammalian cochlear amplification, allowing humans to detect sound pressure as low as 20 μPa.

Prestin (SLC26A5) is abundantly expressed in the lateral membrane of OHCs, where it is responsible for electromotility[2,3]. Prestin is unique in that it directly converts changes in the transmembrane electric potential into mechanical displacements. Unlike other SLC26 family members that function as anion transporters[4], prestin's anion transport activity is very low[5], implying that the voltage-driven motor function of prestin has evolved from an anion transport mechanism[6]. The motor activity of prestin coincides with the voltage-induced movement of voltage sensor charges, which manifests as nonlinear electric capacitance (NLC)[7,8]. Intracellular Cl− ion is an essential co-factor of prestin[9–11], but it remains unclear how Cl− is held in prestin and

[1]Department of Biological Sciences, Graduate School of Science, The University of Tokyo, Bunkyo-ku, Tokyo 113-0033, Japan. [2]Department of Otolaryngology —Head and Neck Surgery, Feinberg School of Medicine, Northwestern University, Chicago, IL 60611, USA. [3]Department of Chemistry, Graduate School of Science, Kyoto University, Kitashirakawa, Oiwake-cho, Sakyo-ku, Kyoto 606-8502, Japan. [4]The Hugh Knowles Center for Clinical and Basic Science in Hearing and Its Disorders, Northwestern University, Evanston, IL 60608, USA. [5]Present address: Department of Life Sciences, Graduate School of Arts and Sciences, The University of Tokyo; Meguro-ku, Tokyo 153-8503, Japan. [6]Present address: MRC Laboratory of Molecular Biology, Cambridge, UK. [7]Present address: Graduate School of Medical Life Science, Yokohama City University, Yokohama, Japan. ✉e-mail: t-2438@yokohama-cu.ac.jp; k-homma@northwestern.edu; nureki@bs.s.u-tokyo.ac.jp

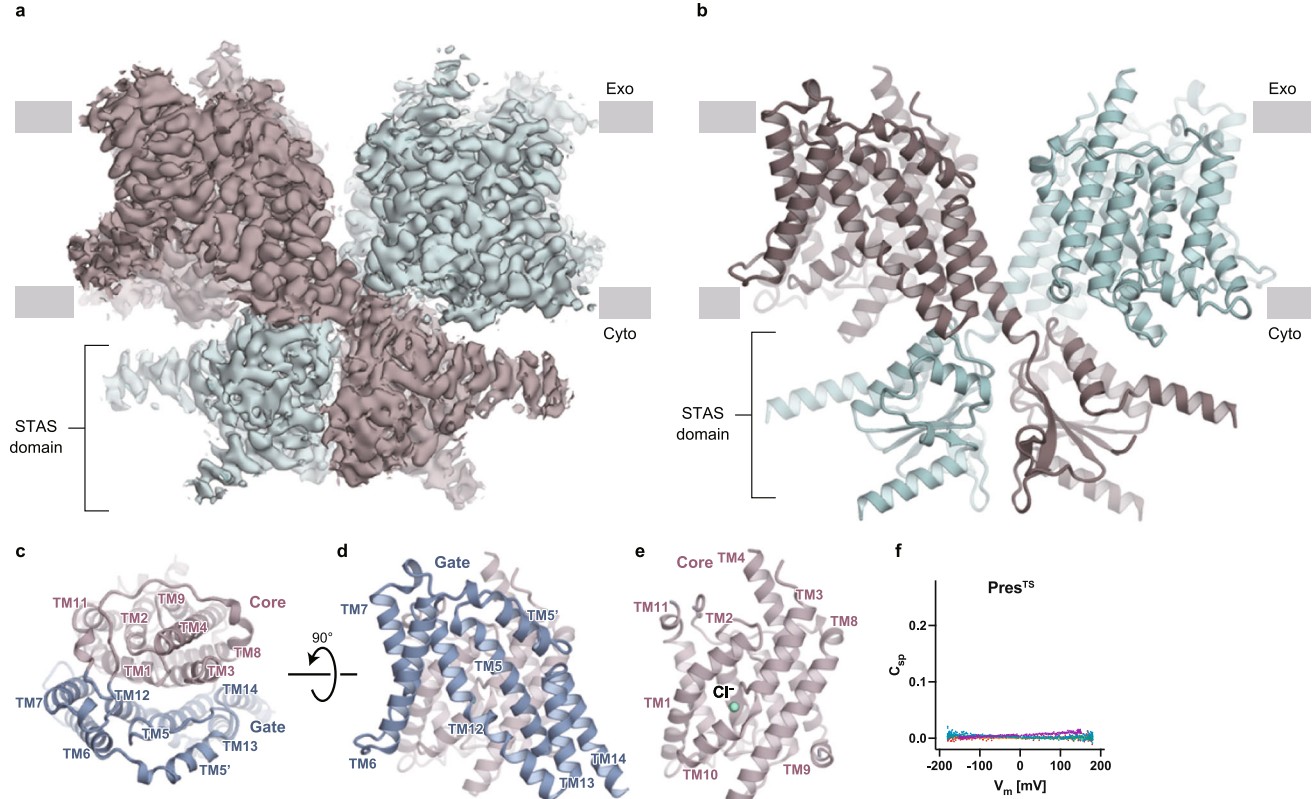

**Fig. 1 | Cryo-EM structure of Pres^TS. a** A Cryo-EM map of the Pres^TS homodimer. Each protomer is shown in a different color (green and dark brown). Gray boxes indicate the location of the lipid bilayer, with the extracellular (Exo) and cytosolic (Cyto) sides shown. **b** Ribbon representation of the Pres^TS homodimer, with protomers shown in the same colors used in panel **a**. **c** Transmembrane domain of a Pres^TS protomer, viewed from the extracellular side. The core (pink) and gate (blue) domains are shown as ribbon models. "TM" indicates transmembrane helix. See Supplementary Fig. 2a for the detailed TM numbering information. **d**, **e** Side views of the gate (**d**) and core (**e**) domains. The green sphere indicates a bound Cl⁻ ion. **f** Voltage-dependent cell membrane electric capacitance measured in HEK293T cells expressing Pres^TS. NLC was undetectable in cells expressing Pres^TS, although it targets the cell membrane (Supplementary Fig. 8). Source data are provided as a Source Data file.

contributes to motor function. Salicylate inhibits the motor activity of prestin[9], which accounts for the reversible hearing loss caused by aspirin overdosage[12]. Previous structures of bacterial SLC26[13], mouse SLC26A9[14], and human SLC26A9[15] have allowed homology modeling of prestin with high confidence, and in addition, very recent structures of prestin from human[16] and dolphin[17] have revealed its molecular structure and Cl⁻-induced conformational change, which advanced our understanding of the voltage-driven motor mechanism of prestin. However, many questions still remain regarding the anion-induced conformational changes.

Here, we present the structures of thermostabilized prestin in complex with Cl⁻, $SO_4^{2-}$, and the prestin inhibitor, salicylate, determined by single-particle analyses using cryo-electron microscopy (EM). These structures, along with recently solved cryo-EM prestin structures[16–18], provide significant mechanistic insights toward the clarification of the voltage-driven motor function of prestin, which is responsible for the exquisite sensitivity and frequency selectivity of mammalian hearing.

## Results

### Structure determination

We performed single-particle analyses of prestin proteins by cryo-EM to elucidate the voltage-dependent motor mechanism of prestin. First, we expressed human prestin (hPres) with FLAG and GFP tags at the C-terminus in human embryonic kidney-293S (HEK293S) cells, and after solubilization by digitonin, the protein was purified by affinity chromatography using FLAG antibody and GFP nanobody-immobilized resins. The size-exclusion chromatography (SEC)

analysis showed a broad peak corresponding to hPres, probably due to its partial aggregation during purification (Supplementary Fig. 1a, b). Consistently, the cryo-EM analysis revealed the highly heterogeneous particle shape and size of the purified hPres (Supplementary Fig. 1c), which provided only poor maps that were not sufficient to define the molecular architecture of hPres. To improve the structural stability, we engineered the thermostabilized Prestin (Pres^TS) by replacing multiple amino-acid residues with those evolutionarily conserved among the SLC26 family[19,20] (Supplementary Fig. 2a, b, Supplementary Data 1). Briefly, we used the HMM search to obtain 10,718 full/partial sequences, including other SLC26 transporters in eukaryotes, and using the hPres sequence as the template, the amino acids conserved in more than a certain percentage of species were left in the original hPres sequence, and less-conserved amino acids were replaced with those best conserved in other species (Supplementary Data 2). Several conservation thresholds from 60–95% were set, and the construct with "60% conserved sequence" showed the most improved stability (Supplementary Fig. 3). The residues that had been presumed to be important for the anion-binding site were then reverted to those of wild-type hPres, and the resultant construct was named as Pres^TS. This strategy allowed the purification of Pres^TS in digitonin micelles with high homogeneity, as indicated by the excellent SEC elution profile (Supplementary Fig. 1d, e). The cryo-EM single-particle analysis yielded a cryo-EM map of Pres^TS at the overall resolution of 3.63 Å (Fig. 1a–e), according to the gold-standard Fourier shell correlation (FSC 0.143) criterion (Supplementary Fig. 4, Supplementary Table 1).

The introduction of the TS mutations resulted in the loss of NLC within the experimentally measurable voltage range (±200 mV)

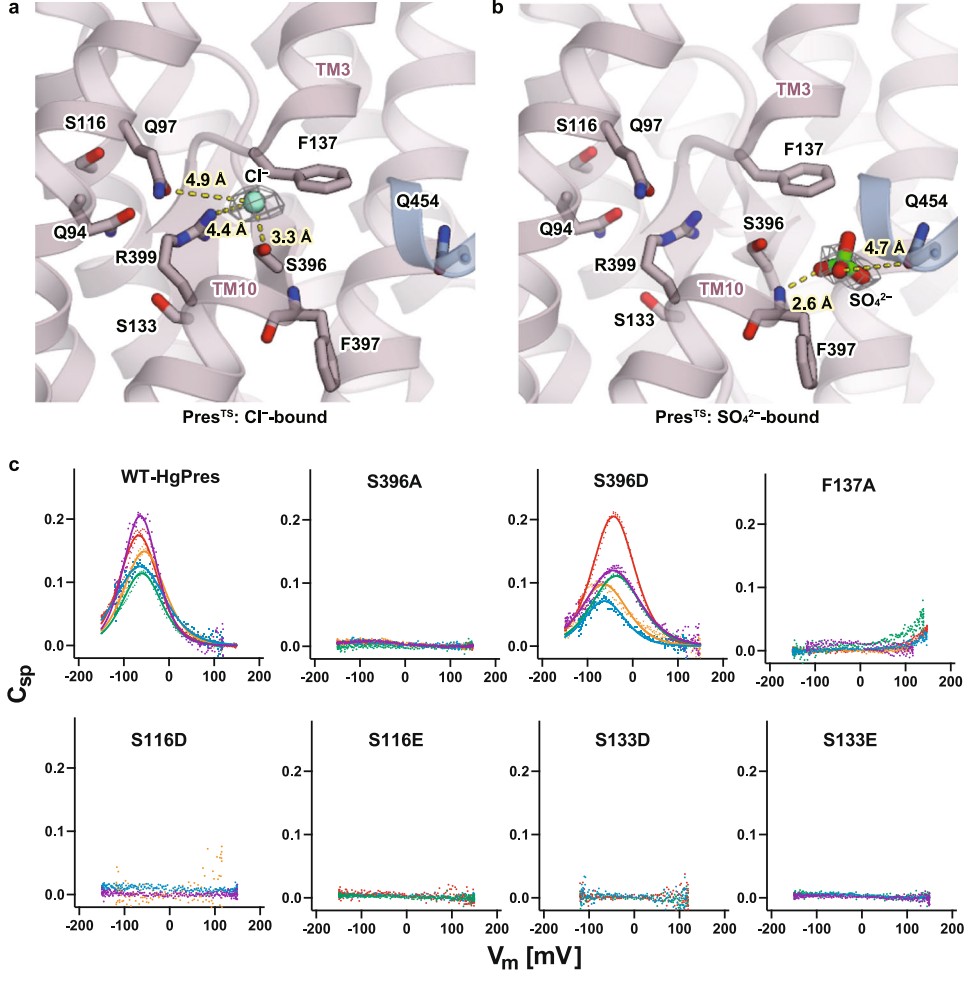

**Fig. 2 | The anion-binding sites of Pres^TS. a, b** Close-up views of the central anion-binding pocket found in the Cl^−-bound (**a**) and SO_4^2−-bound (**b**) Pres^TS structures. Cl^− and SO_4^2− ions are indicated as a green sphere and a stick model, respectively. Residues that contribute to anion binding are also shown with stick models. The cryo-EM densities of the bound anions are indicated by gray meshes. Predicted short-distance interactions are indicated by yellow dotted lines with the distance. **c** Cell membrane electric capacitance measurement in HEK293T cells expressing WT-, S396A-, S396D-, F137A-, S116D-, S116E-, S133D-, or S133E-HgPres. Three to five examples in different colors are shown for each panel. Solid lines for WT- and S396D-HgPres indicate two-state Boltzmann fittings. The $\alpha$, $V_{pk}$, and charge density values (mean ± S.D.) were as follows: [$0.032 \pm 0.006$ mV$^{-1}$, $-65 \pm 14$ mV, and $12 \pm 5$ fC/pF] for WT ($n = 22$); and [$0.029 \pm 0.005$ mV$^{-1}$, $-51 \pm 11$ mV, and $14 \pm 7$ fC/pF] for S396D ($n = 7$). These NLC parameters were statistically indistinguishable between WT vs. S396D (determined by one-way ANOVA followed by Dunnett's multi-comparison tests). Source data are provided as a Source Data file.

(Fig. 1f). Partial reversion of the TS mutations did not rescue NLC, and structural determinations of the 'partially reverted' Pres^TS constructs were unsuccessful.

## Overall structure
The structure of Pres^TS protein forms a domain-swapped homodimer, with each protomer consisting of 14 transmembrane (TM) α-helices and the C-terminal cytosolic Sulfate Transporter and Anti-Sigma factor antagonist (STAS) domain crossing each other. Both the TM and C-terminal domains contribute to dimer formation (Fig. 1a, b). The TM domain is divided into two subdomains: the core (TM helices 1–4 and 8–11) and the gate (TM helices 5–7 and 12–14) domains (Fig. 1c, d). The interdomain interface contains mostly hydrophobic residues. These structural observations indicate that the TM domain of the prestin protomer adopts the typical inverted repeat transporter fold[14,15,21,22], which presently adopts an inward-open conformation. The overall structure remarkably resembles the cryo-EM structures of the mouse and human SLC26A9 anion transporters (Supplementary Fig. 7)[14,15], although these two SLC26 family members have distinct physiological functions. SLC26A9 mediates channel-like fast anion transport[23,24], whereas prestin

(SLC26A5) has voltage-driven motor activity with barely detectable transport activity.

## Anion-binding site
The substrate transport by the SLC26 transporters is considered to follow an alternate access mechanism, in which the substrate binding site is alternately opened toward either side of the membrane, and passively driven by the elevator-like motions of the core and gate domains[14,15]. By analogy to other similarly folded transporters such as UraA[22], band 3 (AE1)[21], SLC26Dg[13], SLC26A9[14,15], and Sultr[25], and also as structurally and functionally investigated in prestin[16–18,26], the substrate binding site is presumably located at the pseudo-symmetric center of the core domain. The amino-terminal ends of the symmetrically arranged unwound helices (TM3 and TM10) are oriented toward this site, creating an electrically favorable environment for capturing negatively charged substrates between the dipoles (Figs. 1e and 2a). In fact, we observed a density at this site, probably corresponding to the Cl^− ion stabilized in this position by R399 and S396 (Fig. 2a). It is likely that F137 contributes to confining the Cl^− ion at the site (Fig. 2a). R399 is a highly conserved residue within the SLC26 family, and its critical roles in anion binding and NLC have been extensively studied[26,27]. To

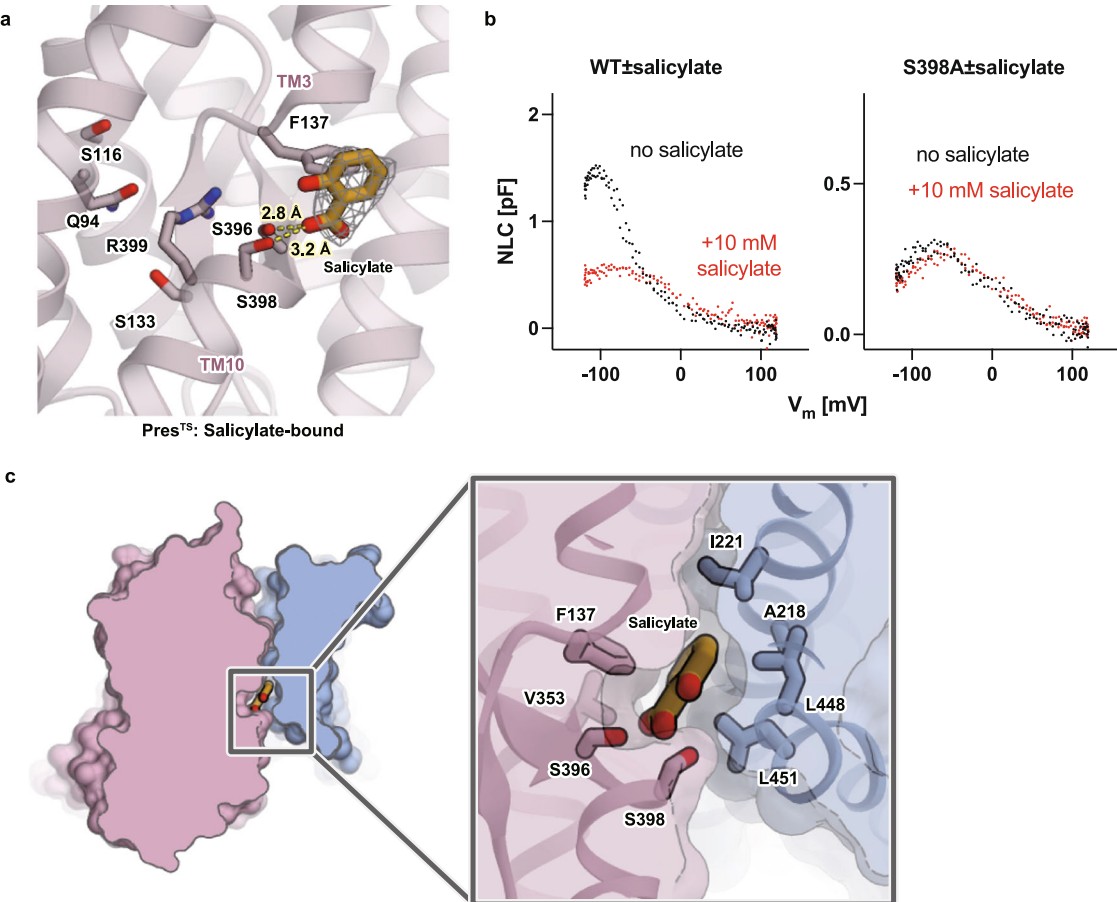

**Fig. 3 | Salicylate binding. a** A close-up view of the central anion-binding pocket of Pres[TS] with a bound salicylate. Salicylate and some important residues are shown in stick representations. The cryo-EM density of salicylate is shown as a gray mesh. Predicted short-distance interactions are indicated by yellow dotted lines with the distance. **b** The effect of salicylate on the NLC of WT (left) and S398A-HgPres (right). NLC was measured from the same cells before (black) and after (red) the application of 10 mM salicylate in the bath solution. The total numbers of recordings were seven for both WT- and S398A-HgPres. Source data are provided as a Source Data file. **c** Salicylate binding pocket. A close-up view of the salicylate binding pocket between the core and the gate domains of Pres[TS] (left panel). Salicylate is sandwiched between the core and the gate domains of Pres[TS] (right panel).

investigate the functional roles of the surrounding residues at the Cl⁻-binding site, we mutated them in naked mole-rat prestin (HgPres) and measured their NLC (Fig. 2c). S396A abolished NLC, while F137A largely shifted the voltage operating point of NLC to the depolarizing direction. These results affirmed the contributions of these residues in creating a favorable environment for anion binding and their importance for the generation of NLC. On the other hand, S396D showed wild-type-like NLC, as reported previously[28,29], probably because the negative charge of aspartate mimics Cl⁻ binding. We also examined the roles of S116 and S133, which are located near R399 but on the opposite side of the anion-binding pocket. S133 is proximal to R399, while S116 is rather apart but connected to R399 through Q94. These residues are likely to form electrostatic interactions with R399. Mutations of these residues to glutamate or aspartate (i.e., S116D, S116E, S133D, and S133E) abolished NLC (Fig. 2c). The nullified NLC of these mutants is not due to impaired membrane targeting, as a membrane-impermeable fluorescent dye successfully labeled these HgPres mutants (Supplementary Fig. 8). These results suggest that NLC requires anion binding at the central pocket in which R399 plays a critical role and that several other residues surrounding this region also contribute to create an optimal environment for anion binding.

To examine anion binding to the central pocket, we performed 10 independent MD simulations for 1 μsec each (10 μs in total) starting from different initial conformations in the inward-open Cl⁻-bound state (Supplementary Figs. 9 and 10. See Methods for details). In 5 of

the 10 simulations, the Cl⁻ ion remained within the pocket for 1 μs. In the remaining five simulations, the Cl⁻ ion was released after 845, 643, 465, 153, and 61 ns, but three of them then showed the spontaneous binding of another Cl⁻ from the bulk solvent (Supplementary Fig. 9), as observed in a previous MD simulation[16]. Overall, these results support the stable binding of the Cl⁻ ion within the pocket. Notably, the coordination distance between the Cl⁻ ion, and R399 found in our cryo-EM Pres[TS] structure is 4.49 Å, which is longer than the typical coordination distances of Cl⁻ ions found in Cl⁻-bound protein structures deposited in the PDB server[30]. A longer coordination distance is also observed in the MD simulation (Supplementary Fig. 9). We then analyzed the contribution of the R399 side-chain and the dipole moments of TM3 and TM10 to the Cl⁻ binding, by calculating their electrostatic interactions with the bound Cl⁻ ion for the MD trajectories (Supplementary Fig. 11). The result revealed that the contribution from the helix dipole moments is larger than that by R399 (details in the legend), explaining the large coordination distance of the Cl⁻ ion in the cryo-EM structure and in the simulations.

It is well known that the replacement of Cl⁻ with $SO_4^{2-}$ largely shifts the voltage operating point ($V_{pk}$) of prestin towards the depolarizing direction[11,31]. Therefore, we next performed a cryo-EM analysis of Pres[TS] in the presence of sodium sulfate (Supplementary Fig. 5). The overall structure of $SO_4^{2-}$-bound Pres[TS] was essentially the same as the Cl⁻-bound form of Pres[TS]. In the presence of 10 mM sodium $SO_4^{2-}$, the density at the pseudo-symmetric center becomes weak, despite the

presence of Cl⁻ ions at the same concentration (300 mM), and a density instead appears nearby, where the solvent is more accessible, and the F397 amide and Q454 side-chain could coordinate an ion (Fig. 2a and b). We supposed that $SO_4^{2-}$ competes with Cl⁻ but is accommodated in the anion-binding site in a slightly different manner, and thus modeled a sulfate ion onto this density. It is conceivable that the known dependence of NLC on various anion species[9,11,31] can be partly ascribed to subtle differences in the binding positions among anions.

### Inhibition by salicylate

Salicylate is a well-known inhibitor of OHC electromotility[32–34], which directly acts on prestin[9]. It is likely that the salicylate-bound state mimics an elongated state[35] because the application of salicylate induces reversible elongation of OHCs along the longitudinal direction (Supplementary Movies 2 and 3, Supplementary Fig. 12a). To understand the mechanism of prestin inhibition by salicylate, we performed a cryo-EM analysis of PresTS in the presence of 30 mM sodium salicylate (Supplementary Fig. 6). The dissociation constants ($K_d$) of salicylate to digitonin detergent-extracted hPres and PresTS were determined by isothermal titration calorimetry (ITC) to be 7.08 and 12.1 mM, respectively (Supplementary Fig. 13a, b), suggesting that the TS mutations did not severely affect salicylate binding and that >70% of the anion-binding site was occupied by salicylate under our experimental condition. We observed a flat-shaped density at the interface between the core and gate domains, which likely corresponds to salicylate (Fig. 3a). The binding position of the polar moiety of salicylate is similar to that of $SO_4^{2-}$, while the benzene ring moiety interacts hydrophobically with F137 and V353 in the core domain, which likely inhibits the relative motion of the core/gate domains and freezes the structure in a certain state (Fig. 3a). In the gate domain, the benzene ring of the bound salicylate fits into the hydrophobic pocket constituted by A218, I(V)221, L448, and L(M)451 (the residues of hPres are indicated in parentheses) (Fig. 3c), in agreement with the mode of salicylate binding to hPres reported in a recent study. These salicylate binding modes are similar to those found in the hPres and dolphin prestin (dPres) structures reported recently (Supplementary Fig. 14f–i). The carboxylate moiety of salicylate was found in the proximity of S396 and S398, implying the contributions of these small polar residues for salicylate binding. Conversions of these residues to acidic ones (e.g., S396D/E or S398D/E) reportedly made prestin constitutively active and insensitive to inhibition by salicylate[28,29]. We confirmed that prestinS396D and prestinS396E are indeed insensitive to salicylate (Supplementary Fig. 12b). These mutations may neutralize the positive charge of R399 and likely mimic the anion binding and thus prevent salicylate binding. We found that S398A also makes prestin insensitive to salicylate, further affirming the importance of the small polar residue for salicylate binding (Fig. 3b, Supplementary Fig. 12b).

### Interdomain interaction between the core and the gate domains

The recently solved structures of hPres, dPres, and gerbil prestin (gPres) suggested that electromotility generation and concurrent NLC are intimately related to the relative rigid-body displacement between the core and gate domains[16–18], which is analogous to the elevator-like motion of transporters[14,15,36]. We also observed a small relative rigid-body displacement between the core and gate domains, as deduced from our Cl⁻ vs. salicylate- bound PresTS structures (Supplementary Fig. 16a and Supplementary Movie 1), which is consistent with the presumed elevator-like motion[14,15,36]. However, structural comparisons with the wild-type hPres[16] revealed notable differences between hPres and PresTS, especially for the Cl⁻-bound state: PresTS adopts an inward-open-like state, whereas hPres adopts an occluded-like intermediate conformation (Fig. 4a–c, d-left and Supplementary Fig. 16a). In contrast, in the salicylate-bound state, both hPres and PresTS adopt similar conformations (an intermediate state) (Fig. 4d right, Supplementary Fig. 16b), consistent with the observation that the mode of salicylate

binding is similar between PresTS and hPres (Fig. 3, Supplementary Fig. 14f–i). These results suggest that PresTS is stabilized in an inward-open conformation that was not captured in recent studies by others.

We then investigated the interdomain interactions that are responsible for the conformational change. The Q97, N447, and E293 residues participate in hydrogen-bonding interactions between the gate and core domains (Cl⁻-bound PresTS) (Fig. 5a), while this interdomain interaction collapses upon the transition toward the Cl⁻-occluded state as captured in the recent prestin structures (Supplementary Fig. 15a). Q97 and N447 are located near the Cl⁻ binding site, and their mutations, Q97D and N447D, almost completely abolished NLC (Fig. 5d). In addition, we examined the mutations of the charged residue E293, which is farther from the Cl⁻ binding site and forms a hydrogen bond with N477. A previous study showed that E293Q abolishes NLC[37]. The N447A and E293A mutations retained NLC, but the magnitudes were greatly reduced. The charge-reversing mutations, N447D and E293K, completely abolished NLC. These results suggest that the domain movement upon NLC generation is highly associated with the interdomain hydrogen bond interactions, and their modulation by anion binding plays critical roles in domain movements and, thus, in NLC generation. Distinct intra- and interdomain hydrogen-bonding networks revealed by a comparison of the recently reported prestin structures, (Supplementary Fig. 15) further suggest their dynamic roles and contributions to NLC-inducing conformational changes. We also examined the role of a hydrophobic residue, V353, which is near the anion-binding site and is exposed to the core/gate domain interface. The conversion of this hydrophobic residue to hydrophilic ones (V353D, V353E, and V353N) drastically reduced or abolished NLC (Fig. 5d). Substitutions to other hydrophobic residues only had mild effects on NLC when the size was smaller (V353A, V353I, and V353L), but bulky residues (V353F, V353Y, and V353W) decreased or abolished NLC (Fig. 5d). These results suggest that the hydrophobic interactions, mediated by these residues, are also important to facilitate the presumed elevator-like motion and thus for NLC generation.

### Dimerization of prestin

A pivotal hydrophobic interface formed by V499 and I500 (mutated to leucine in PresTS) is located at the C-terminus of the long TM14 helix and involved in the domain-swapped dimerization interface of prestin (Fig. 6a and Supplementary Fig. 16c). Mutations that attenuate the hydrophobicity of these residues, such as V499G and I500G, almost completely abolished NLC (Fig. 6b), in agreement with previous reports[38,39]. Even a modest mutant (I500L) notably affected the voltage sensitivity ($0.032 \pm 0.006$ mV⁻¹ vs. $0.017 \pm 0.001$ mV⁻¹, $p < 0.0001$) and voltage operating point ($-65 \pm 14$ mV vs. $-22 \pm 8.5$ mV, $p < 0.0001$) of prestin (WT, $n = 22$ vs. I500L, $n = 5$, mean ± SD) (Fig. 6b), as in the V499A mutant of mouse prestin reported previously[38]. Although similar mutants such as V499I and I500A have only minor effects on NLC (Fig. 6b), it seems that the stable domain-swapped dimer formation is crucial for NLC generation, and changes in the dimer interface could affect NLC. It is likely that an elevator-like motion of the core domain underlies NLC, and the rigid dimerization of the gate domains facilitates this conformational transition. We also observed strong densities of lipid acyl chains and cholesterols on the gate domains (Fig. 6c), further suggesting the importance of their stabilization. Prestin activity (electromotility and concurrent NLC) is sensitively modulated by the lipid composition and thickness of the cell membrane[40–42]. For example, the voltage operating point of prestin is tuned by the cholesterol contents[40]. Therefore, the lipid molecules observed on the TMD surface and at the dimer interface provide significant insights and suggest their direct involvement in and contribution to the molecular conformational changes of prestin.

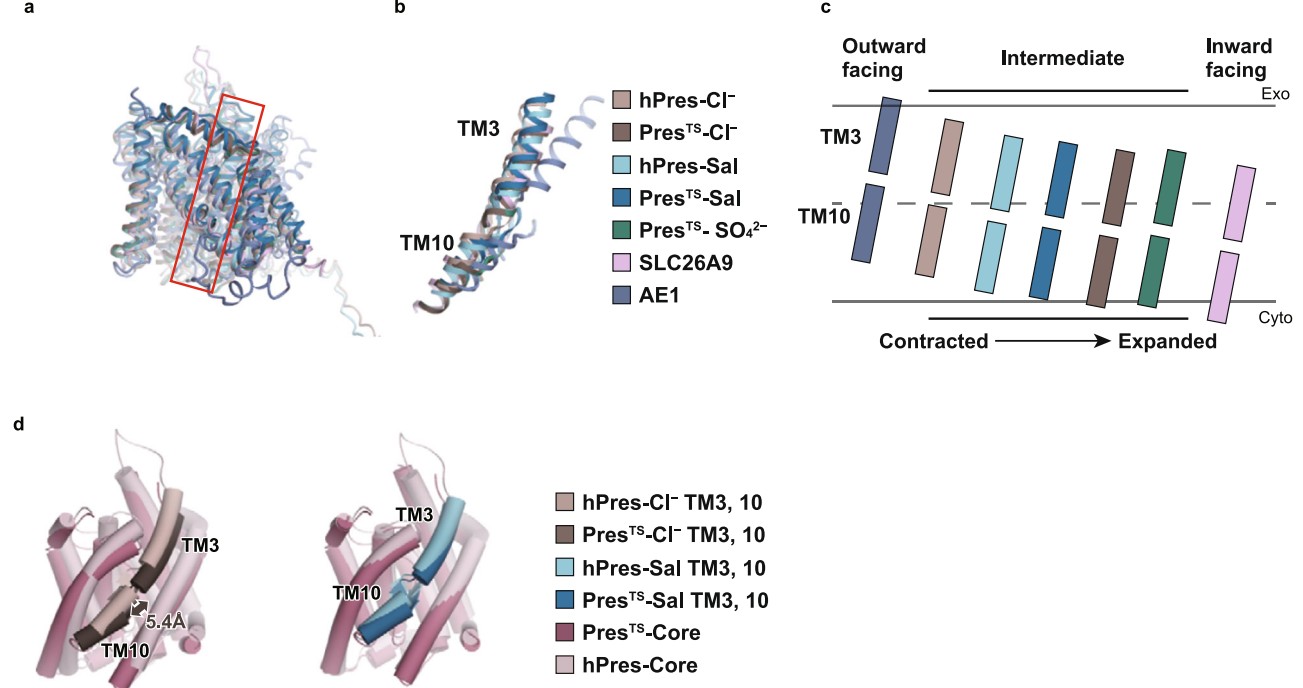

**Fig. 4 | Comparisons of prestin structures. a** Structural comparison of hPres-Cl⁻ (PDB ID: 7LGU), Pres^TS-Cl⁻, hPres-Sal (PDB ID: 7LH2), Pres^TS-Sal, Pres^TS-SO₄²⁻, SLC26A9 (PDB ID: 6RTC), and AE1 (PDB ID: 4YZF). The models are superimposed with respect to the gate domain. The red box indicates the location of TM3 and 10. **b** Comparison of the structures of TMs 3 and 10 with those from other species (**a**). **c** Schematic diagram showing the relative positions of TM3 and TM10 within the membrane found in the prestin structures reported herein and by others. The structures are superimposed at the gate domain, and the TM3/10 positions are illustrated based on the centrally located S398. **d** Structural comparison of Pres^TS and hPres. The models are superimposed at the gate domain, and the TM3/10 positions are illustrated based on the centrally located S398. The Cl⁻-bound (left panel) and salicylate-bound (right panel) states are shown. The displacement distance of TM3/10 between Pres^TS and hPres was determined using S398 (in TM10) as the reference point. The structures are superimposed at the gate domain, and the TM3/10 positions are illustrated based on the centrally located S398.

## Discussion

In this study, we revealed the structures of thermostabilized Pres^TS in inward-open conformations. Although SLC26 transporters are thought to alternate between inward- and outward-open conformations, mammalian prestin orthologs may not assume an outward-open conformation[43]. However, it is conceivable that the rigid-body motions of the core and gate domains, which would be similar to the elevator-like motions presumed for other SLC26 anion transporters[14,15,27], occur in a voltage-dependent manner in prestin (Fig. 7), and that such conformational changes are intimately linked to the electromotility and concurrent NLC. R399 and the symmetrically arranged N-termini of the TM3 and TM10 helices provide a strong positively charged electric field that attracts anions at the center of the core domain (Fig. 2). In good agreement, our MD simulations showed stable accommodation of the Cl⁻ ion in the positively charged cavity, as well as its spontaneous binding from the bulk solvent (Supplementary Figs. 9 and 10). It is unlikely that such a highly charged cavity is isolated from the bulk solvent without being neutralized by a counter charge (Fig. 7a). Therefore, anion binding to the positively charged cavity within the prestin protein would be essential for the transition from the inward-open state to the occluded state, like hPres in the Cl⁻-bound conformation (Fig. 7a, b). Mutations that result in the neutralization of this positively charged local environment, such as S396D and S396E, likely mimic Cl⁻-bound states, thereby allowing the conformational transitions of prestin. According to the MD simulations, the unusually long coordinate distance is ascribed to the additional positive electrical fields created by the helix dipoles of TM3 and TM10 (Supplementary Fig. 11). Salicylate binds to this positively charged area with its carboxylate mimicking the anionic moiety, but the aromatic benzene ring is sandwiched between the core and the gate domains, thereby

sterically restricting the elevator-like motion of the core/gate domains (Figs. 3c and 5c). These observations suggest that the bound anion and surrounding environment are likely to serve as the extrinsic voltage sensor in prestin, and their longitudinal migration due to the elevator-like movement might be detected as NLC[9,44], while other charged residues in the core domain may also contribute to the net charge movement[37], as proposed based on recent structures[16,17]. We can further speculate that membrane depolarization decreases negative charges on the membrane inner surface, readily attracting Cl⁻ ion to the inner surface and facilitating its translocation to the positively charged anion-binding pocket constituted by TM3 and TM10 dipoles and R399, enabling the rigid-body movement of the core domain with respect to the gate domain.

The results from our mutational analyses, which focused on the residues located at the gate/core domain interface (Fig. 5), suggested the presence of a hydrogen-bonding network that affects the putative voltage-driven, relative rigid-body motions between the two domains. It is structurally possible for anion binding to affect this hydrogen-bonding network. We found that the Cl⁻ and SO₄²⁻ binding sites are distinct (Fig. 2a, b), probably due to the differences in their sizes and charge valences. It is thus likely that the hydrogen-bonding network, which probably affects the elevator-like movement of the core/gate domains, is distinctly affected by different anions, thereby accounting for the known anion dependence of NLC. Collectively, we propose the dual functional significance of anion binding to prestin for its voltage-driven motor function. Firstly, neutralization of the positively charged central cavity by anion binding is essential for allowing prestin to transit to the putative occluded state (Fig. 7b). Secondly, modulation of the hydrogen-bonding network by a bound anion is requisite for the rigid-body motions of the core domain with respect to the gate

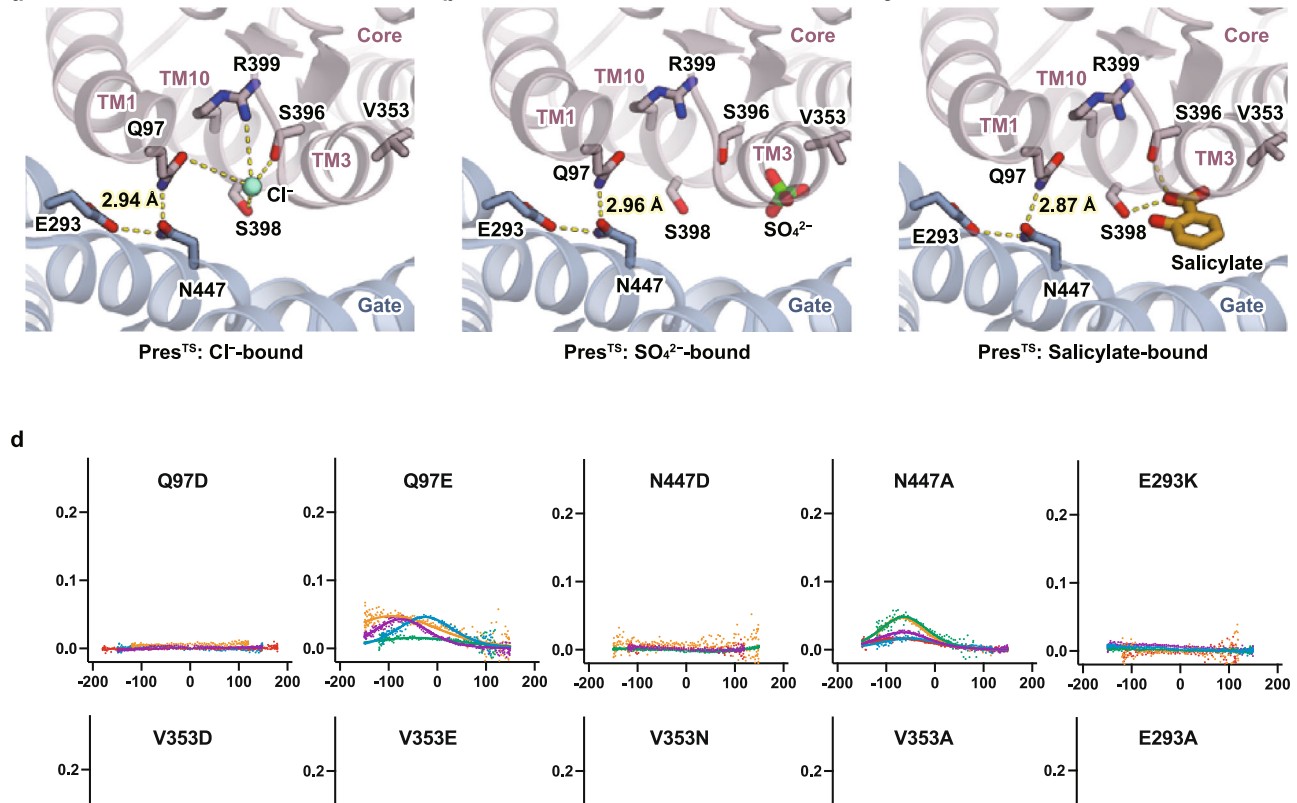

**Fig. 5 | Interaction between the core and gate domains. a–c** The core-gate domain interface in the Cl⁻-bound (**a**), SO₄²⁻-bound (**b**), and salicylate-bound (**c**) Pres$^{TS}$ structures, viewed from the extracellular side. Putative hydrogen bonds are indicated by yellow dotted lines. **d** Cell membrane electric capacitance measurement in HEK293T cells expressing Q97D-, Q97E-, N447D-, N447A-, E293K-, E293A-, V353D-, V353E, V353N-, V353A-, V353I-, V353L, V353Y, V353W, and V353F-HgPres. Four to five examples in different colors are shown for each panel. Solid lines (for Q97E-, E293A-, N447A-, V353A-, V353L-, V353I-, V353F-, and V353Y-HgPres) indicate two-state Boltzmann fittings. The α, $V_{pk}$, and charge density values (mean ± S.D.) were as follows: [0.021 ± 0.006 mV⁻¹, −59 ± 30 mV, 8 ± 3 fC/pF] for Q97E ($n = 4$); [0.023 ± 0.007 mV⁻¹, −63 ± 33 mV, 7 ± 7 fC/pF] for E293A ($n = 10$); [0.027 ± 0.004 mV⁻¹, −65 ± 8 mV, 4 ± 2 fC/pF] for N447A ($n = 5$); [0.036 ± 0.004 mV⁻¹, 51 ± 8 mV, 8 ± 4 fC/pF] for V353A ($n = 6$); [0.034 ± 0.011 mV⁻¹, −92 ± 18 mV, 10 ± 6 fC/pF] for V353L ($n = 6$); [0.031 ± 0.004 mV⁻¹, −63 ± 20 mV, 8 ± 4 fC/pF] for V353I ($n = 6$); and [0.022 ± 0.003 mV⁻¹, 89 ± 6 mV, 13 ± 5 fC/pF] for V353F ($n = 6$). Source data are provided as a Source Data file.

domain. These two effects cooperatively allow the voltage-dependent conformational change of the core domain of prestin. Since prestin is highly unlikely to function without an extrinsic anion, the possibility that the bound anion also contributes to voltage sensing[9,44] should not be disregarded.

The voltage-dependent conformational changes of prestin molecules are likely to be directly converted into OHC contraction and elongation, which can be up to ~5% of the total length of an OHC[45,46]. We speculate that the putative voltage-induced elevator-like motions of the core domains underlie this electromechanical conversion. We also surmise that rigid dimer formation is important for the voltage-driven motor function of prestin. The cryo-EM structure of Pres$^{TS}$

suggested highly stable dimer formation, mediated by hydrophobic residues at the C-terminal ends of TM14 (Fig. 6a), and multiple lipids filling the dimeric junction region (Fig. 6c). In fact, the V499G and I500G mutations, which are expected to destabilize the dimeric interaction between prestin protomers, abolished NLC (Fig. 6b). Rigid dimerization-assisted fixation of the orientation of the gate domains, with respect to the axis perpendicular to the membrane plane, might be crucial for making the yet to be identified voltage sensor of prestin optimally responsive to changes in the transmembrane electric potential, and thus for efficiently converting the motions of the core domains into molecular motility. The elevator-like motions of the core domains with respect to the homodimeric gate domains with two-fold

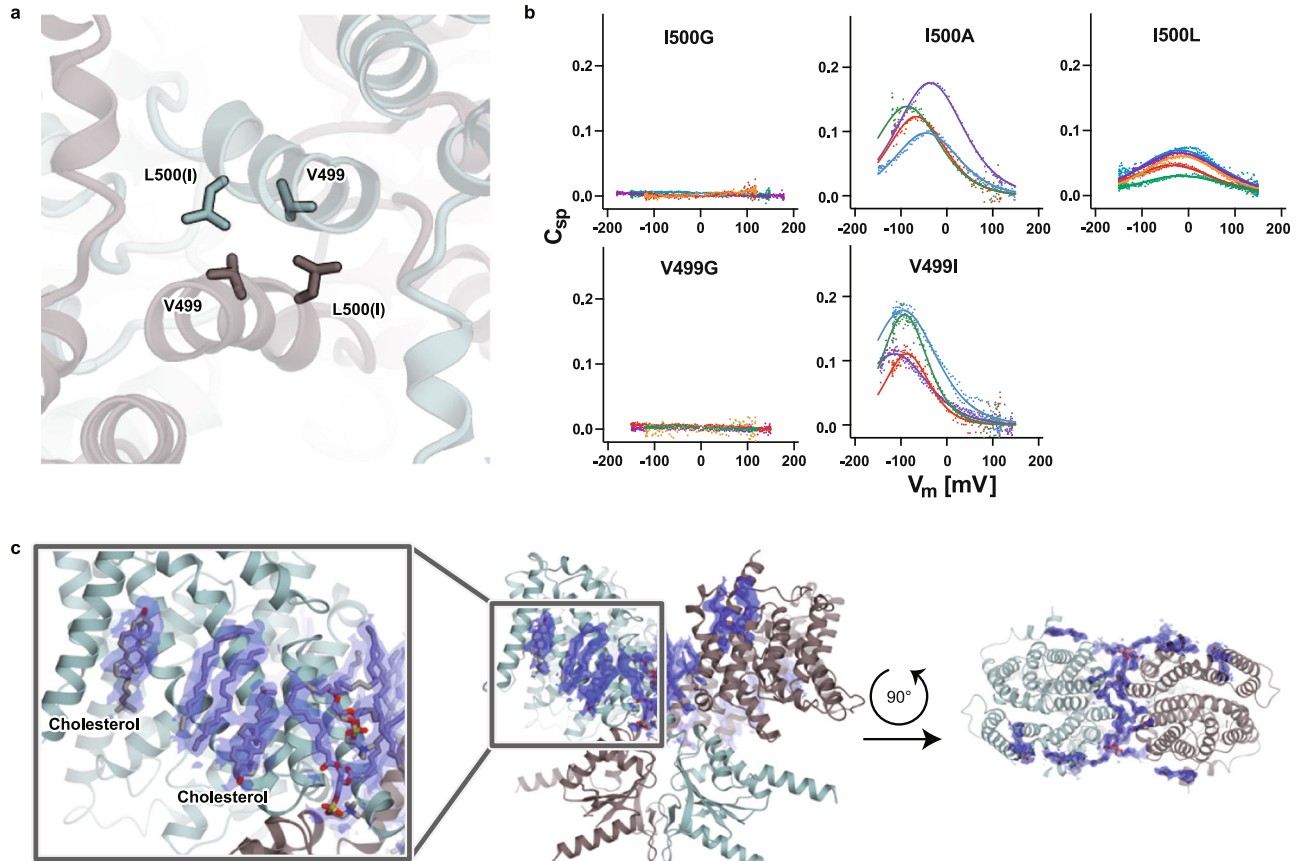

**Fig. 6 | Dimerization of Pres^TS. a** Hydrophobic interactions are mediated by V499 and L500 (I500 in wild-type) at the C-termini of the TM14 helices. **b** Cell membrane electric capacitance measurements in HEK293T cells expressing I500G-, I500A-, I500L-, V499G-, and V499I-HgPres. Five examples in different colors are shown for each panel. Solid lines (for V499I-, I500A-, and I500L-HgPres) indicate two-state Boltzmann fittings. The α, $V_{pk}$, and charge density values (mean ± S.D.) were as follows: [0.027 ± 0.008 mV$^{-1}$, −104 ± 21 mV, 23 ± 14 fC/pF] for V499I ($n$ = 6); [0.025 ± 0.006 mV$^{-1}$, −58 ± 36 mV, 20 ± 9 fC/pF] for I500A ($n$ = 8); and [0.017 ± 0.001 mV$^{-1}$, −22 ± 8 mV, and 13 ± 4 fC/pF] for I500L ($n$ = 5). Source data are provided as a Source Data file. **c** Lipids found on the Pres^TS structure. Phospholipids and cholesterols are shown as gray stick models with cryo-EM densities (blue). A close-up view of lipids bound to Pres^TS (left panel). A lateral view (center panel) and an extracellular view (right panel) of lipids found on Pres^TS.

symmetry probably produce the directional motor activity. Therefore, understanding how prestin molecules are connected to the submembrane structure is crucial, in order to fully appreciate the molecular and cellular mechanisms of OHC electromotility responsible for the exquisite sensitivity and frequency selectivity of mammalian hearing.

In summary, the cryo-EM structures of the thermostabilized prestin, Pres^TS, obtained under various ionic conditions and by MD simulations, showed the unique anion binding by prestin. Extensive functional analyses of HgPres mutants have revealed the critical residues within the prestin protein for the NLC generation. Our results suggest that anion binding neutralizes the positively charged anion-binding site and modulates the interdomain hydrogen-bonding interactions. This allows the elevator-like motion of the core domain with respect to the gate domains that are stabilized by the rigid dimer architecture, leading to robust expression of electromotility. The present study thus provides important mechanistic insights into the prestin-mediated conversion of electric energy into mechanical work.

## Methods
### Plasmids preparation
Full-length human prestin (hPres) was amplified using human universal reference cDNA (Zyagen) as the template and cloned into a modified pEG BacMam vector with a C-terminal TEV cleavage site followed by a FLAG-EGFP tag. A thermostabilized prestin protein (Pres^TS) was designed by the consensus mutagenesis approach, based on the

sequences of all eukaryotic prestin orthologs (https://github.com/TaizoAyase/consensus_creator)[20]. Briefly, we used the HMM search to obtain 10,718 eukaryotic sequences, including duplicated and/or partial sequences, for the alignment. Using the hPres sequence as the template, the amino acids conserved in more than a certain percentage of species were left in their original sequence, and nonconserved amino acids were replaced with those best conserved in other species, while the residues near the anion-binding site were kept as wild-type. Several conservation thresholds from 60 to 95% were set, and the construct with "60% conserved sequence" showed the most improved stability (Supplementary Fig. 3a). The residues considered to be important for the anion-binding site were then reverted to those of wild-type hPres, and the resultant construct was named as Pres^TS. A cDNA encoding Pres^TS was synthesized with codons optimized for expression in human cell lines, and cloned into the modified pEG BacMam vector described above. For functional assays, naked mole-rat (*Heterocephalus glaber*) prestin (HgPres) was used for better expression as compared to hPres. The amino-acid sequence of HgPres is 96% identical to that of hPres (Supplementary Fig. 2). Wild-type HgPres cDNA was synthesized based on the mRNA sequence (XM_004839711.2) by General Biosystems (Durham, NC), and cloned into the pSBtet-Pur vector[47] with a C-terminal mTurquoise2 (mTq2) tag. Standard mutagenesis methods were used to generate HgPres mutants in the pSBtet-Pur vector. The primer used in this study is listed in the source data file.

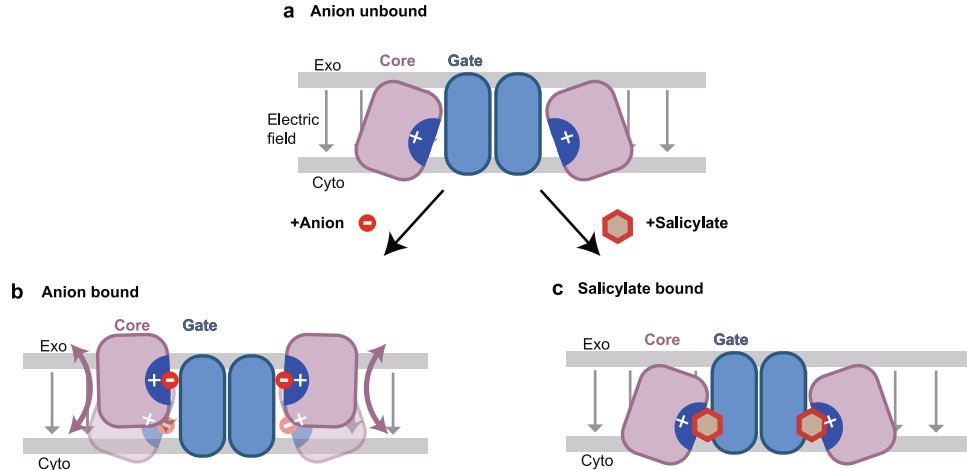

**a** Anion unbound

**b** Anion bound

**c** Salicylate bound

**Fig. 7 | Hypothetical elevator-like movement of the core domains facilitated or inhibited by anion binding.** Prestin dimers with the core domain ("Core", purple) and the gate domain ("Gate", blue) are shown with the plasma membrane (gray lines), with the extracellular (Exo) and cytosolic (Cyto) sides indicated. Gray arrows indicate the electric field. **a** Unbound state. The highly positively charged central pocket (indicated by "+") is forcibly exposed toward the intracellular solvent in the absence of an anion. **b** Anion binding allows the core domain to adopt the occluded state. Pink arrows indicate the elevator-like motion of the core domain movement with respect to the gate domain. Anion binding neutralizes the positive charge at the anion-binding pocket macroscopically. However, microscopic charge biases likely remain, which may serve as the voltage sensor of prestin. **c** Salicylate binding to the positively charged central pocket, but sterically inhibits the transition from the inward-open to the occluded state.

## Fluorescence-detection size-exclusion chromatography-based thermostability (FSEC-TS) assay

The FSEC-TS[48,49] analysis was conducted to assess the thermal stability and to screen constructs for structural analysis. HEK293S GnTI⁻ cells (ATCC, CRL-3022) were grown and maintained in FreeStyle 293 medium (Gibco) at 37 °C with 8% $CO_2$. For the FSEC-TS protein expression, transient transfection was performed. The pEG BacMam vector plasmid (2 µg) was added to a 10 mL culture of HEK293S GnTI⁻ cells, at a density of $0.5 \times 10^6$ cells/mL, with 5 µL of Lipofectamine 3000 (Invitrogen), 4 µL of P3000 (Invitrogen) and 200 µL of Opti-MEM™ I Reduced Serum Medium (Invitrogen). After 48 h, the cells were collected by centrifugation ($5000 \times g$, 12 min, 4 °C) and washed twice with PBS. The cells were then solubilized in buffer containing 50 mM Tris-HCl, pH 8.0, 300 mM NaCl, 10% glycerol, 2 mM 2-mercaptoethanol (β-ME), and 1% digitonin (Calbiochem), for 1 h at 4 °C. After ultracentrifugation ($186,000 \times g$, 20 min, 4 °C), the supernatant was collected and heat-shocked with a thermal cycler for 10 minutes in a temperature range from 4 to 90 °C. The supernatant was then loaded on a Superose 6 Increase 10/300 GL column (GE Healthcare) attached to a fluorescence detector to monitor GFP fluorescence. The column was equilibrated in buffer containing 50 mM Tris-HCl (pH 8.0), 150 mM NaCl, 0.03% DDM, and 0.006% CHS.

## Expression and purification of hPres and Pres^TS

Baculoviruses carrying the prestin constructs were produced and amplified in Sf9 cells, using the Bac-to-Bac system (Invitrogen). HEK293S GnTI⁻ cells (ATCC, CRL-3022) were grown and maintained in FreeStyle 293 medium (Gibco) at 37 °C with 8% $CO_2$. For protein expression, the baculovirus was added at 1/10 (v/v) to the culture medium of HEK293S GnTI⁻ cells, at a density of $3 \times 10^6$ cells/mL. After 16–18 h, 5 mM valproic acid was added, and the cells were further incubated at 30 °C with 8% $CO_2$ for 48 h. The cells were collected by centrifugation ($5000 \times g$, 12 min, 4 °C) and lysed by sonication in buffer, containing 50 mM Tris-HCl, pH 8.0, 300 mM NaCl, and protease inhibitors (1.7 µg/mL aprotinin, 0.6 µg/mL leupeptin, 0.5 µg/mL pepstatin and 1 mM PMSF). After cell debris removal by centrifugation ($4000 \times g$, 12 min, 4 °C), the membrane fraction was collected by ultracentrifugation ($186,000 \times g$, 1 h, 4 °C). The membrane fraction was solubilized in buffer, containing 50 mM Tris-HCl, pH 8.0, 300 mM NaCl, 10% glycerol, 2 mM 2-mercaptoethanol (β-ME), and 1% digitonin (Calbiochem), for 1 hr at 4 °C. After ultracentrifugation ($186,000 \times g$, 20 min, 4 °C), the supernatant was collected and incubated with anti-FLAG M2 affinity gel (Sigma) for 1 h at 4 °C. The resin was washed with 10 column volumes of wash buffer, containing 50 mM Tris-HCl, pH 8.0, 300 mM NaCl, 20% glycerol, 2 mM β-ME, and 0.1% digitonin. Bound prestin was eluted with the wash buffer containing 0.125 mg/mL of FLAG peptide, and EGFP fluorescence-positive elution fractions were collected. For further purification, CNBr-Activated Sepharose 4 Fast Flow Beads (GE Healthcare) conjugated with anti-GFP nanobodies[50] were added to the collected fractions and incubated for 1 h at 4 °C. The resin was washed with 10 column volumes of the same wash buffer and then gently suspended with TEV protease (purified in-house) overnight at 4 °C to cleave the C-terminal FLAG-EGFP tag. For this digestion, 1 mg of TEV protease was added per 10 mg of protein. After the TEV protease cleavage, the flow-through fraction was pooled, concentrated to 5–10 mg/mL using a centrifugal filter device (Millipore 100 kDa MW cutoff), and loaded onto a Superose 6 Increase 10/300 GL column (GE Healthcare), equilibrated in buffer containing 50 mM Tris-HCl, pH 8.0, 300 mM NaCl, 2 mM β-ME, and 0.1% digitonin. The peak fractions were pooled and concentrated to 5–10 mg/mL.

## Electron microscopy sample preparation

After precipitating aggregated proteins by ultracentrifugation at $138,000 \times g$ for 20 min, a 3 µL portion of the supernatant was spotted onto a glow-discharged holey carbon grid (Quantifoil R1.2/1.3, Cu/Rh, 300 mesh), which was plunge-frozen in liquid ethane using a Vitrobot Mark IV (FEI) at 6 °C with a blotting time of 4 sec with 100% humidity. To obtain $SO_4^{2-}$- and salicylate-bound Pres^TS, the purified protein solution was mixed with sodium sulfate (10 mM) or sodium salicylate (30 mM), respectively, and incubated on ice for 15 min before grid preparation.

## Electron microscopy data collection and processing

The prepared grids were transferred to a Titan Krios G4 microscope (Thermo Fischer Scientific), equipped with a Gatan Quantum-LS Energy Filter (GIF) and a Gatan K3 Summit direct electron detector. The camera was operated in the correlated double sampling (CDS) mode. The dataset was collected at a nominal magnification of ×105,000, corresponding to a calibrated pixel size of 0.83 Å per pixel (The University of Tokyo, Japan). Each movie was recorded for

5.0 seconds and subdivided into 64 frames. The electron flux rate was set to 7.5 e⁻/pix/s at the detector, resulting in an accumulated exposure of 54 e⁻/Å$^2$ at the specimen. The data were automatically acquired using the SerialEM 3.7.4 software[51], with a defocus range of −0.8 to −1.6 μm.

Initially, all datasets were corrected for beam-induced motion, using the motion correction program implemented in the single-particle analysis software RELION-3.1[52], and the contrast transfer function (CTF) parameters were estimated using CTFFIND4.1.13[53]. For the dataset of the Cl⁻-bound state, 1,373,022 particles were picked from 4680 micrographs by using the Laplacian-of-Gaussian picking function in RELION-3.1, and were used to generate two-dimensional (2D) models for reference-based particle picking. Particles were extracted with down-sampling to a pixel size of 3.63 Å/pix and subjected to several rounds of 2D and 3D classifications. The best class contained 341,744 particles, which were then re-extracted with a pixel size of 1.10 Å/pix and subjected to 3D refinement. The resulting 3D model and particle set were subjected to per-particle CTF refinement, beam-tilt refinement, Bayesian polishing[54], and 3D refinement. The final 3D refinement and postprocessing of the three classes yielded maps with global resolutions of 3.63 Å, according to the FSC = 0.143 criterion[55]. The local resolution was estimated using RELION-3.1. The processing strategy is described in Supplementary Fig. 4. For the dataset of the SO$_4$$^{2-}$-bound state, 1,181,278 particles were picked from 4077 micrographs, and the best class containing 249,144 particles was selected and processed as above, yielding a map with a global resolution of 3.52 Å (Supplementary Fig. 5). For the dataset of the salicylate-bound state, 848,704 particles were picked from the 3375 micrographs by reference-based particle picking with the map of the Cl⁻-bound state. The best class containing 113,410 particles was selected and processed, yielding a map with a global resolution of 3.57 Å (Supplementary Fig. 6).

### Model building and validation
The models of the Cl⁻-bound state of Pres$^{TS}$ were manually built de novo in the Cryo-EM density map in COOT-0.9.6[56], facilitated by the previously reported crystal structures of SLC26Dg (PDB ID: 5DA0). The final model includes residues T13–K580 and Y614–S725, while the omitted loops are disordered and not visible in the cryo-EM maps. Structure refinement was initially performed with Rosetta[57] and phenix.real_space_refine from PHENIX-1.14-3260[58,59]. After manual adjustments, the models were then subjected to structure refinement with the Servalcat pipeline using REFMAC5[60,61] and manual real-space refinement in COOT-0.9.6[56]. The models of the salicylate-bound state were built by using the Cl⁻-bound model as the starting model. The statistics of the 3D reconstruction and model refinement are summarized in Supplementary Data Table 1. All molecular graphics figures were prepared with CueMol 2.2.3.443 (http://www.cuemol.org) or ChimeraX 1.3[62] (https://www.rbvi.ucsf.edu/chimerax/).

### NLC measurement
HEK293T-based stable cell lines expressing HgPres-mTq2 constructs in a doxycycline-dependent manner were established and maintained in DMEM (11965, Thermo Fisher Scientific), supplemented with 10% FBS and 1 μg/ml puromycin (A11138, Fisher Scientific), as previously described[63,64]. The expression of the HgPres constructs was induced by adding 1 μg/mL doxycycline hyclate (D9891, Sigma) to the culture medium, one day prior to NLC recording. Whole-cell NLC recordings were performed at room temperature, using an Axopatch 200B amplifier (Molecular Devices, Sunnyvale, CA). Recording pipettes were pulled from borosilicate glass to achieve initial bath resistances averaging 3–4 MΩ. Recordings were performed using sinusoidal (2.5 Hz, 120–150 mV amplitude) voltage stimuli superimposed with two sinusoidal stimuli (390.6 (f1) and 781.3 (f2) Hz, 10 mV amplitude).

Recording pipettes were filled with an intracellular solution containing 140 mM CsCl, 2 mM MgCl$_2$, 10 mM EGTA, and 10 mM HEPES (pH 7.3). Cells were bathed in an extracellular solution containing 120 mM NaCl, 20 mM TEA-Cl, 2 mM CoCl$_2$, 2 mM MgCl$_2$, and 10 mM HEPES (pH 7.3). Osmolality was adjusted to 304 mOsmol/kg with glucose. Current data were collected and analyzed by jClamp 23.1.1 (SciSoft Company, New Haven, CT)[65].

NLC (C$_m$) data were analyzed using the following equation:

$$C_m = \frac{\alpha Q_{\max} \exp\left[\alpha\left(V_m - V_{pk}\right)\right]}{\left\{1 + \exp\left[\alpha\left(V_m - V_{pk}\right)\right]\right\}^2} + C_{\text{lin}} \tag{1}$$

where α is the slope factor, Q$_{max}$ is the maximum charge transfer, V$_m$ is the membrane potential, V$_{pk}$ is the voltage at which the maximum charge movement is attained, and C$_{lin}$ is the linear capacitance. The magnitude of NLC (C$_m$ − C$_{lin}$) was corrected for cell size (C$_{lin}$) because larger cells tend to express greater amounts of the prestin protein (C$_{sp}$ ≡ (C$_m$ − C$_{lin}$)/C$_{lin}$). Prism version 8.0.0 (GraphPad Software, www.graphpad.com) was used for the curve-fitting analysis of both motility and NLC.

### Cell imaging
Stable cell lines carrying mTq2, mTq2-tagged Pres$^{TS}$, and HgPres variants were seeded on cover glasses, and after the addition of 1 μg/mL doxycycline (0.1 μg/ml for mTq2 alone control cells), they were cultured for two days. Cells were then rinsed once in PBS, fixed in 4% PFA for 10 min at room temperature, rinsed again in PBS, and mounted on slides, using DAKO fluorescence mounting media (S3023, Dako). Cell images were captured using Nikon A1R+ confocal laser microscope systems with the same laser power settings, except for the mTq2 alone control cells and the Pres$^{TS}$ cells, where lower laser power settings were used to avoid image saturation.

### Quantification of protein expression
Stable cell lines carrying mTq2, mTq2-tagged Pres$^{TS}$, and HgPres variants were seeded on 12-well plates and treated with 1 μg/ml doxycycline (0.1 μg/ml for mTq2 alone control cells) for 1–2 days. Cells were collected in Cell Dissociation Buffer (Thermo Fisher 13150016), resuspended in PBS, and dispensed into a 96-well plate. mTq2 fluorescence intensities and absorbance at 660 nm were measured using a Synergy Neo2 plate reader. mTq2 fluorescence values were divided by OD$_{660nm}$ to determine expression levels.

### Cell surface protein labeling and quantitation
Stable cells were seeded on a six-well plate, and the expression of mTq2-tagged HgPres constructs was induced by 3 μg/mL doxycycline for 2 days prior to labeling. Cells were washed once with PBS, and 2 mL of 10 μM Sulfo-Cyanine3 NHS ester (Lumiprobe) dissolved in ice-cold PBS was added (per well) and incubated for 30 minutes at 4 °C. The reaction was stopped by the addition of 200 μL of 100 mM glycine. Cells were collected and lysed by sonication on ice in 500 μL of lysis buffer (150 mM NaCl, 20 mM HEPES, pH 7.5, 1 mM EDTA, 20 mM DDM, 1 mM DTT, and 50 μg/mL leupeptin). The lysate was centrifuged at 16,000 × g for 5 min at 4 °C. A GFP selector slurry (5 μL, NanoTag Biotechnologies) was added to the supernatant and incubated for 30 minutes at 4 °C, with end-over-end mixing using a rotator. Bound proteins were collected alongside the GFP selector by brief centrifugation, and observed with a fluorescent microscope (Leica DMIRB) controlled by μManager[66]. Merged images of GFP selectors in cyan and red channels were analyzed using FIJI[67] to determine the fluorescent signal intensities of mTq2 and Cy3. The results are shown in Supplementary Fig. 8.

## Isothermal titration calorimetry analysis

Salicylate binding to prestin molecules was measured using a MicroCal ITC 2000 microcalorimeter (GE Healthcare) at 20 °C. The Pres$^{TS}$ and hPres proteins were purified as described above. The peak fractions from the SEC were collected and diluted to 0.02–0.04 mM as a monomer (2–3 mg/mL) with the SEC buffer. The ligand solutions used for titration were prepared by adding sodium salicylate to the SEC buffer, at a final concentration of 200 mM, to include the concentration for grid preparation. The ligands were injected 20 times (0.4 μl for injection 1, 2 μl for injections 2–20), with 150 s intervals between the injections. The background data obtained from the buffer sample were subtracted before the data analysis. Thermograms were integrated and baseline corrected with the Origin7 software package (MicroCal). The peak integration data were analyzed with the pytc software package[68] to perform the single-site binding model fitting and parameter estimation. The measurements were repeated twice, and similar results were obtained.

## Animals

The care and use of the animals in this study were approved by the National Institutes of Health and the Animal Care and Use Committee at Northwestern University. Mice were group housed with food and water provided ad libitum under a 12 h light/12 h dark cycle and temperatures of 18–23 °C with 40–60% humidity. OHCs were isolated from wild-type FVB/NJ mice as described previously[38].

## Molecular dynamics simulation

We performed molecular dynamics (MD) simulations to analyze interactions of a Cl$^-$ ion binding to prestin. Wild-type hPres model was constructed by modeler[69], using the Cl$^-$-bound Pres$^{TS}$ cryo-EM structure as the template. The simulation system includes a monomer part of the transmembrane domain consisting of residues R58 to P506 (449 residues). Hydrogen atoms were added using the psfgen plugin in NAMD[70]. The protonation states of ionizable residues were assumed to be those at pH 7 by PROPKA[71]. The structure of prestin was manually embedded in a 105 × 105 Å 1-palmitoy-2-oleoyl-sn-glyceroo-3-phosphocholine (POPC) bilayer modeled with VMD[72] for LINUXAMD64 version 1.9.3. The bilayer was solvated with TIP3P water molecules with Na$^+$ and Cl$^-$ ions of 0.15 M using the solvate plugin and the autoionize plugin in VMD for LINUXAMD64 version 1.9.3. The simulation box was initially 105 × 105 × 120 Å in size and contained 110,295 atoms.

We first performed a preparatory MD simulation with the NAMD 2.12b1 program package. The force field parameters of the protein, water molecules, and lipids were CHARMM27[73], TIP3P[74,75], and CHARMM36[76], respectively. Short-range nonbonded interactions were cut off at 12 Å with a force-switching function. Long-range electrostatic interactions under periodic boundary conditions were calculated with the particle mesh Ewald method. Temperature and pressure were controlled with Langevin dynamics[77] and the Langevin piston method[78], respectively. In the regulation of the pressure, the x- and y-axes were isotropically scaled (NPTiso), where the z-axis was perpendicular to the membrane surface. The initial degrees of freedom of water molecules were constrained with the SETTLE[79] algorithm. Bonds, including hydrogen atoms, were constrained with the RATTLE[80] method.

We first performed a 1000-step energy minimization followed by a 160 ps heating to 200 K under the NVT conditions, and a 40 ps heating simulation to 300 K under the NPTiso conditions, with only the protein, water molecules, and lipid tails being allowed to move. After a 10 ns equilibrium MD simulation under NPTiso conditions, with only the water molecules and lipid tails being allowed to move, a 10 ns equilibrium MD simulation under NPTiso conditions without any restraints was performed. The force field parameters were then switched to those of the Amber force field[81] (ff14SB, lipid17, and TIP3P for

the protein, lipids, and water molecules, respectively), and a 50 ns equilibrium simulation was carried out with NAMD 2.12b1 for Linux-x86_64-MP.

From the last snapshot of the equilibrated MD system obtained by the preparatory simulation described above, an MD simulation with distance restraints for the bound Cl$^-$ ion at the binding site was performed for 500 ns with pmemd.cuda.MPI of the AMBER ver. 16 program package[82–84] with the Amber force fields to sample the initial conformation of the system for long-time production runs. The distance restraints for Cl$^-$ were applied with two restraint functions of distances from S396:O$_\gamma$ and R399:C$_\zeta$ to keep the Cl$^-$ ion in the binding pocket. From the MD simulation with the restraints, 10 initial conformations were sampled every 50 ns. Finally, 10 independent MD simulations without restraints for 1 μs each (10 μs in total) were performed from the initial conformations. We sampled 1000 conformations from each MD trajectory for 1 μs (10,000 conformations in total).

We calculated the RMSDs of the protein (Supplementary Fig. 10) by CPPTRAJ[85] and confirmed that the protein structures were stable during the trajectory calculations. We classified the protein conformations into the Cl$^-$ bound one and the Cl$^-$ unbound one with three distances of the nearest Cl$^-$ ion from R399-C$_\zeta$, S396-O$_\gamma$, and S398-O$_\gamma$, $r_1$, $r_2$, and $r_3$, respectively (Supplementary Fig. 9). The protein conformation was classified into the bound one when $4\,\text{Å} < r_1 < 10\,\text{Å}$, $2\,\text{Å} < r_2 < 6\,\text{Å}$, and $2\,\text{Å} < r_3 < 6\,\text{Å}$, and into the unbound one when $r_1 > 15\,\text{Å}$, $r_2 > 10\,\text{Å}$, and $r_3 > 10\,\text{Å}$. Among the 10,000 overall conformations in the 10 μs MD simulations, 8281 conformations (corresponding to 8.281 μs) and 1497 conformations (corresponding to 1.497 μs) were classified into the bound and unbound ones, respectively. We calculated the averaged electrostatic interactions of the bound Cl$^-$ ion with the protein in the sampled Cl$^-$ bound conformations. The same point charges of the protein atoms as those used in the MD simulations were employed for the electrostatic calculations. Electrostatic energy contributions from the main-chain dipoles of the individual amino acids (Supplementary Fig. 11a–c) and those from the entire individual amino acids (Supplementary Fig. 11d, e) were computed by direct sums of the Coulombic interactions between the group considered without periodic images of the periodic boundary condition. The relative electric permittivity of the Coulombic interactions was set to 1. The electrostatic calculations were carried out with an in-house program.

### Reporting summary

Further information on research design is available in the Nature Research Reporting Summary linked to this article.

## Data availability

The data that support this study are available from the corresponding authors upon request. Cryo-EM density maps obtained in this study have been deposited in the Electron Microscopy Data Bank under the accession codes EMD-31757 (Cl$^-$-bound), 31758 (SO$_4^{2-}$-bound), and 31759 (salicylate-bound). Atomic coordinates obtained in this study have been deposited in the Protein Data Bank under IDs 7V73 (Cl$^-$-bound), 7V74 (SO$_4^{2-}$-bound), and 7V75 (salicylate-bound). The raw images have been deposited in the Electron Microscopy Public Image Archive, under accession code EMPIAR-11199. Source data are provided with this paper.

## Code availability

Consensus sequence creator script code used in this study is available at Zenodo 7099885. An in-house program for electrostatic calculations are available from the corresponding authors upon request.

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

## Acknowledgements

We thank the staff scientists at The University of Tokyo's cryo-EM facility, especially K. Kobayashi, H. Yanagisawa, A. Tsutsumi, M. Kikkawa, and R. Danev, and computational time at the Research Center for Computational Science, Okazaki, Japan. Cell imaging work was performed at the Northwestern University Center for Advanced Microscopy. Funded by: The MEXT Grant-in-Aid for Specially Promoted Research 16H06294 and JST CREST program 20344981 (to O.N.), JSPS KAKENHI grant 17H05000 and 20H03216 (to T.N.), AMED JP19am0101115 (support number 1111), NIH grant DC017482 (to K.H.), JSPS KAKENHI grant 19H03195 and 20H05441 (to S.H.). Northwestern University Center for Advanced Microscopy is supported by NCI CCSG P30 CA060553 awarded to the Robert H Lurie Comprehensive Cancer Center.

## Author contributions

H.F. planned the experiments. H.F. prepared the cryo-EM samples. K.H. and S.T. performed the functional analyses. H.F. and T.N. collected and processed the cryo-EM data and built the structures. K.Y. and A.T. assisted with data processing and structure refinement. T.S. and S.H.

performed the MD simulation. H.F. drafted the original manuscript. H.F., S.H., T.N., K.H., and O.N. substantially contributed to the revision of the manuscript drafts. M.F., R.U., T.K., T.N., K.H., and O.N. supervised the research.

## Competing interests

O.N. is a cofounder and scientific advisor for Curreio. The remaining authors declare no competing interests.
