## [Peer Review File · Nature Communications]

Cryo-EM structures of thermostabilized prestin provide mechanistic insights underlying outer hair cell electromotilityREVIEWER COMMENTS

Reviewer #1 (Remarks to the Author):

This manuscript is the fourth in a burst of papers describing cryo-EM structures of SCL26A5, or prestin, the piezoelectric-like actuator protein of auditory outer hair cells that drives cochlear amplification in mammals. Experimental structures of this protein have been awaited eagerly, to provide the basis for understanding the seemingly unique electromechanical behavior.

The structures presented are technically sound and contribute to a structural framework for further analysis of molecular mechanisms.

However, given the mentioned parallel reports, the message is largely confirmatory (maybe with the exception of a sulfate bound structure). The same is true for the larger part of the accompanying functional results reported here, that to a substantial part replicate or extend previously published data.

The approach taken by Fukamata et al, differs from the 3 competitive papers in that they determine the structure of a mutationally modified ('thermostabilized') version of prestin, apparently helping to obtain a well-resolved cryo-EM structure. In contrast, the competitors achieve comparable data quality with native prestin isoforms.

Unfortunately, this approach entails the major limitation of the present study as this construct lacks function, i.e. the non-linear charge movement (NLC) thought to reflect the molecular transitions underlying electromotility. Consequently, the obtained structures are highly unlikely to reveal the dynamic behavior of prestin. In fact, the presented structures bound to different anions, which based on functional data are known to drive changes in state occupancy, show the same 'frozen' conformation. In contrast, structures with chloride, sulfate and salicylate obtained by Ge et al (2021, Cell) and Bavi et al. (bioRxiv, 2021) differ from each other and reveal dynamic behavior of prestin.

Nevertheless, given the near-synchronous emergence of these data with the mentioned companion papers, as well as the importance of these structures, this report is certainly worth publishing.

However two major aspects need to be changed :

1. The authors suggest throughout the manuscript that their data show an elevator-like/rigid-body movement of prestin. However, their data do not provide any additional evidence to such a mechanism, as the structures are essentially identical ('frozen construct') and the functional (mutational) experiments that probe the interface are consistent with dynamics between core and gate domains but cannot specifically show an elevator-like movement. Clearly, the manuscript's title 'reveal the electromotility mechanism' is not supported by the data. Therefore this claim needs to be toned down massively, including the summary scheme in Fig.5. (This said, such a mechanism appears quite likely,

particularly in the light of some of the competitors' data, but the results in this manuscript provide no substantial evidence beyond what is already known.)

2. A comparison to the abovementioned published structures in Ge et al. (Cell, 2021), Bavi et al. (bioRxiv, 2021), and Butan et al. (bioRxiv, 2021) is warranted, given that those data and conclusions are available at this time.

Specific points:

3. Logic underlying the generation of 'thermostabilized' prestin (l.79 and Suppl. Fig. 2).

'we engineered the thermostabilized hPres (hPresTS) by replacing multiple amino acid residues with those evolutionally conserved among prestin orthologs'

According to alignment in Fig. S2 there are many replacements that - contradicting this statement - do not introduce conserved amino acids, but residues not found in SLC26A5 orthologs. What is the logic behind this?

4. L 110.f. With respect to the identity of the anion binding site only references to non-SLC26 transporters are made. In fact, binding sites for several SLC26 transporters have been structurally determined (SLC26Dg: Geertsma et al. 2015; SLC26A9: Walter et al., 2019; Sultr: Wang et al., 2021). Functionally, anion binding in SLC26A9 (Walter et al., 2019) and even in prestin itself (Gorbunov et al., 2014) have been analyzed in detail.

5. L. 115 ff., coordination of chloride should be compared to results from GE et al., where R399 seems to be not directly involved.

6. L. 122.; S396D/E : mutants were already reported and characterized elsewhere

7. L.129, Fig. S7: I am not convinced by this method for measuring membrane targeting, although elegant in principle. It can account for the relative percentage of the protein at the PM, but not the overall expression level. In fact, the examples shown in SF7 suggest that the levels may differ substantially which may contribute to effects seen in NLC recordings. Moreover, there is reportedly a strong variability in membrane localization between individual cells, is this the case here? NLC recordings were done 1 day after induction, but PM localization assay after 2 days. Therefore, I would like to see high-quality images of the high PM localization of prestin required for reliably measuring NLC.

8. MD simulations: the position of chloride appears different in the simulation compared to the experimental structure. A quantitative comparison is required. Also, only a single 100ns run of the MD simulation is shown. Were repetitions made?

9. Fig 3c: salicylate effect on overall structure: it is difficult to understand which conformation is the salicylate-bound one (this is also true for the movie), but it seems that it is the one with the core domain slightly shifted outwards? The reorientation needs to be quantified. Even considering the limited informational value of the dysfunctional TS-variant, the observation should suggest that salicylate poises

the proteins towards an outward-open or intermediate state rather than an inward open, as proposed here, which would be consistent with the data of Ge et al.. Please clarify.

10. Lengthening of OHCs with salicylate: l.148 and Suppl. Movie: time course and reversibility need to be shown to make sure that the effect is via prestin and not by disrupting cellular structures as also described previously. Quantification of results (elongation, statistics) is also required.

11. L.177: mutational analysis: results are consistent with the importance of the gate-core interface in prestin function, but certainly do not provide evidence for an elevator-like motion.

Minor points.

12. Fig. 3d-f: helix labeled 'TM10' should rather be TM1 (with Q97), with TM10 located below?

13. Abstract , l.27 'resulting in distinct modulations of nonlinear capacitance (NLC), playing an important role in electromotility.' Sentence is largely incomprehensible; also, falsely suggests that these are results from the presented study.

Reviewer #2 (Remarks to the Author):

Please see report in attached PDF.

Futamata et al. present the cryo-EM structures of a thermostabilized version of prestin, the cochlear amplifier, with bound chloride, sulfate and salicylate anions. Mechanistic insights into prestin function are provided. The contribution is important, but suffers from the very recent publication by Ge et al. (2021), Cell from the Gouaux group.

Major points:

- Lines 269-270: Sentence “About 40% of the amino acid residues were altered in hPresTS as compared to wild-type hPres (Supplementary Fig. 2).”

Question of principle: Can a protein be named “hPres” having only about 60% identity to human prestin? The reviewer suggest to rename the protein to “PresTS” not “hPresTS”, because of the important difference in identity.

- Please indicate in a Table or list (in Supplementary Information), which mutations were introduced into human prestin to generate hPresTS: From Supplementary Fig. 2 it is difficult and time-consuming to identify the mutations. This will help the reader to see the difference between wt and TS at a glance.

- Please indicate in the sequence comparison of Supplementary Fig. 2 the degree of similarity and identity, e.g., by introducing the symbols *, . and : below the last sequence (currently hPresTS): you will get this information when performing, e.g., alignment with ClustalW. Again, this is helpful for the reader.

- Lines 124-126: It is unclear to the reviewer why these Ser residues were mutated. Please add more information about the rationale of this experiment. E.g. what are the distances between S116 and S133 to R399? (please include this information): from the Figure it looks like the Ser residues and guanidinium group of R399 are not in distances allowing interactions. Please also elaborate on the discussion of these mutagenesis results – currently unsatisfactory.

- MD simulation experiment in Supplementary Figure 8: Please provide and display at least one additional replica to show reproducibility of these results.

- Figure 2a and b: it seems that the rotamers of S396 are different in the chloride and sulfate bound structures. Please discuss this, e.g., are there new interactions? Please also introduce distance, it is difficult to estimate distances in Figures (general issue). For example, is in Fig. 2b S396 O atom in hydrogen bond-distance to R399 N atoms?

- Figure 2b: Please show the environment around the sulfate ion in a panel or Suppl. Fig. What are the interactions of the protein with the sulfate anion? Please show and discuss.

- Paragraph on ‘Inhibition by salicylate’: What is the binding constant of salicylate to prestin? Any numbers from the literature? Please comment on this, readers will want to have an impression on the affinity of this inhibitor.

- Lines 161-163, statement: “We found that S398A also makes prestin insensitive to salicylate, further affirming the importance of the small polar residue for salicylate binding (Fig. 3b, Supplementary Fig. 9b).” What about S396A? – on Fig. 3a also this residue interacts

with the carboxyl group of salicylate. Please add functional data or reference to literature on the effect of -OH group removal. Please indicate distances in Fig. 3a (S396 and S398 to salicylate).

- Lines 165-166: Rigid body displacement between the core and gate domains: Please quantify this displacement, e.g., indicating distances between the two states, moving angle of a specific transmembrane helix, etc. From this statement the reviewer cannot assess if the displacement is relevant or not.

- Line 170: "Gln97 and Asn447 partly constitute the chloride binding site, and their mutations...". Please show and mention also these residues in Figure 2, where the chloride binding is shown and discussed.

- Last paragraph of "Inhibition by salicylate". If a negative charge is mutated into a positive (E293K), and hydrophobic residues into polar/charged ones (V353D, V353E and V353N), the reviewer is not astonished to see that the function is reduced or abolished. Did the authors perform less radical mutations, e.g., E293A or E293Q, or V353F, V353Y or V353W? Please introduce results into manuscript.

- Paragraph: 'Dimerization of prestin' (page 5). V to G, and I to G are (again) radical mutations, when considering the high flexibility introduced by glycine and the fact that this amino acid is not chiral. Said differently, I am not astonished about the dramatic effect of such mutations on the function. What about the V499A and I500A mutations, i.e., how is the effect on the NLC ?

- Discussion: 4.49 Å for the distance between Arg399 and the chloride ion is relatively large, and uncommon. Did the authors consider the possibility to have water bridging Arg399 and chloride ion? What is the local resolution in this area (area around Arg399 and chloride ion)?

- The density maps are not well documented. Please add in Suppl. Information for all structures: i) Local resolution (2 views, e.g., top and side view) calculated using MonoRes or another program and ii) Euler angle distribution (again two views).

- The rotamer outliers of 7V75 (salicylate) are with 5.7% high (also those from the other two structures are relatively high). Are these outliers in functionally important regions? Please comment.

- The RMSD from the bond angles are quite high. Together with the high % of rotamer outliers, do the authors believe that there is room for improving the model?

- Please provide an SDS-PAGE gel analysis (e.g., in Suppl. Fig. 1) of the protein used for SEC and cryo-EM study. Currently not documented.

- Please expand the discussion, on how the benzene ring should sterically restrict the elevator-like motion of the core/gate domains: from Fig. 3f, 5c, this is difficult to imagine.

- Please add a summarizing last paragraph (in Discussion) or a Conclusion.

- Please provide in Suppl. Information the codon-optimized DNA sequence of hPresTS.
- Subchapter: "Purification of hPres and hPresTS": Please consider that you do not only explain purification here, but also overexpression: Please check title, currently suboptimal.
- You used 10 mM sulfate and 30 mM salicylate for cryo-EM experiments. What are the binding constants of these two anions for hPres and your hPresTS (see also my previous question above about Kd of salicylate)? How many fold excess of ligand (anion) was finally added for grid preparation?
- **Important:** Considering the very recently published structures by Ge et al. (Sept. 2, 2021) Cell from the Gouaux group, please perform a comparison between the structures of this work and Ge et al. (i.e., Pres-Cl, Pres-Sal and Pres-Sulfate). Please incorporate the comparison in manuscript and discuss.

Minor points:

- Introduce the abbreviation STAS for non-expert readers
- Please check Supplementary Fig. 6: there are fine horizontal and vertical gray lines, which in my opinion do not have to be there (probably involuntarily introduced during figure assembly).
- I guess it is more appropriate to write "SLC26" and not "SLC26A", i.e., only "SLC26" if not a specific transporter is addressed, but stated in a general context. Please check.
- Please use one style only, i.e., do not mix one letter and three letter amino acid coding: currently mixed in text and Figures (e.g., also Suppl. Fig. 8a three-letter code).
- Do not mix uncapitalized and capitalized panel labeling, e.g., compare line 222 and line 227.

Reviewer #3 (Remarks to the Author):

Reviewer comments for «Cryo-EM structures reveal the electromotility mechanism of prestin, the cochlear amplifier.

In this study a thermostabilized Prestin variant is studied by cryo-EM and corresponding functional assays. The authors present structures of Prestin bound to chloride, sulfate and salicylate. They identify residues important for binding of chloride and sulfate, and for inhibition by salicylate. The study is nicely carried out, well written and interesting. Their EM data is convincing, and conclusions and speculations made from the structures seems reasonable and are well supported by their NLC data. I still however have some comments that I think should be addressed.

1.

Initial efforts to solve the structure of human Prestin (hPres) by cryo-EM was not successful due to problems in obtaining homogenous protein fractions from size exclusion chromatography. In order to obtain a more stable variant of hPres, using a consensus mutagenesis approach they make a hPres thermostabilized variant hPresTS which they use to solve the chloride, sulfate and salicylate bound structures. In the thermostabilized Prestin variant, approximately 40% of the amino acid residues were altered compared to hPres WT. The thermostabilized Prestin variant has no detectable NLC activity.

When making a thermostabilized Prestin variant, in which 40% of amino acids are mutated to consensus residues, the authors should consider renaming the thermostabilized Prestin variant PresTS instead of hPresTS as the variant is just as much a generic Prestin variant as a human Prestin variant. I also think that the number 40%, or a more precise description of the differences between PresTS and Pres WT should be included in the result section.

It is also stated that partial reversion of the TS mutations neither rescued NLC activity or allowed structural determination, but there is no description of which partially reverted hPres variants that were tested, or how the SEC profile of these variants looked compared to hPresWT and hPresTS. I am also wondering whether the hPres WT had measurable NLC activity or if it was only HgPres that could be used in NLC measurements.

2.

Further, they described the Prestin variant as thermostabilized but does not show, or describe any assay showing that the altered Prestin variant is actually more thermostable. From Figure S1 hPresTS elutes in a nice monodisperse peak (contrary to hPres WT), but additional experiments showing thermostabilization should be performed in order to claim that hPresTS actually is thermostabilized.

3.

Recently 4 Cryo-EM structures (2.3 – 4.3 Å) of Cl⁻ bound and salicylate bound hPrestin was published (Ge et al, Cell, 2021) but there is no reference to these structures or this paper in the manuscript. These Prestin structures have been available online since August 2021 and should be referred to in the introduction.

Further, as these are structures of WT hPrestin, the current manuscript could benefit from a discussion around similarity or differences between the structures of hPresWT and hPresTS.

4.

In the description of the anion binding site Arg399 is pointed out as a key residue in coordination of the chloride ion, but it is not among the residues that are mutated and tested in NLC measurements. Although the relevance of Arg399 in the coordination of Cl seems very likely it would be nice to either include data showing this or cite earlier work describing the importance of this residue.

5.

In the section describing dimerization of Prestin, a figure showing more of the dimerization interface, including also the hydrophilic residues mentioned in the discussion should be included. Are these residues for “stable domain swapped dimer formation” and NLC generation?

Further the relevance of lipid acyl chains and cholesterol (also shown in fig 4C and 4D) on Prestin stabilization and the effects of lipid manipulation on the function of the protein should be explained more thoroughly. To me, these final statements in the result section is a bit cryptic.

7.

Finally I have a few small comments on the methods section:

Line 266: a citation to the used pEG BacMam vector or a description about what modifications that were done to the vector should be included.

Line 301: Could you include the concentration or relative amount of TEV that was used?

Line 347 and 352: From the PDB validation report it seems like parts of the structures could not be modeled. A description of which parts of Prestin that was modelled and which parts that were excluded / could not be modeled due to lack of density in EM data should be included.

Line 397: Was there any missing residues in the structure, between residue 58-506, and where these modeled in or left out from MD simulations?

Reviewer #4 (Remarks to the Author):

This paper presents three cryo-EM structure of human prestin (using a thermostabilized mutant): a Cl⁻-bound structure at 3.52 Å resolution, a sulfate (SO₄²⁻) bound structure (3.52 Å), and a salicylate bound structure (3.57 Å). The thermostabilized protein does not show the NLC (non linear conductance) typical for active prestin so the protein structure is that of a non-functional or inhibited prestin. Functional measurements (NLC) of mutants (of the WT protein) are used together with a short MD simulation to gain insight into the mechanism of prestin.

Overall, the paper falls short in really revealing the electromotility mechanism of prestin (as promised in the title "Cryo-EM structures reveal the electromotility mechanism of prestin, the 2 cochlear amplifier") because the data does not contain evidence of the conformational changes ("elevator like", as described by the authors) that they hypothesize are required for the area changes that undely prestin's electromechanical action in the outer hair cell membrane. The paper presents a number of additional data points, in particular with regard to the potentially important interactions between core and gate domain, that add to the recent papers on prestin, namely the recent cryo-EM prestin structures from human [1], dolphin [2], and gerbil [3].

[1] J. Ge, J. Elferich, S. Dehghani-Ghahnaviyeh, Z. Zhao, M. Meadows, H. von Gersdorff, E. Tajkhorshid, and E. Gouaux, "Molecular mechanism of prestin electromotive signal amplification," *Cell*, vol. 184, no. 18, pp. 4669–4679.e13, 2021.

[2] N. Bavi, M. D. Clark, G. F. Contreras, R. Shen, B. G. Reddy, W. Milewski, and E. Perozo, "Prestin's conformational cycle underlies outer hair cell electromotility," *Nature*, vol. e-pub, 2021.

[3] C. Butan, Q. Song, J.-p. Bai, W. Tan, D. S. Navaratnam, and J. Santos-Sacchi, "Single particle cryo-em structure of the outer hair cell motor protein prestin," *bioRxiv*, 2021.

1) It is always difficult to write a paper when similar work is being published almost at the same time. I would **suggest** to include a comparison to the already published structures and perhaps find an

explanation for why the salicylate bound, thermostabilized structure appears to differ from the salicylate bound structures in [1] and [2] (at least as far as I can tell). This might shed some light on the effect of the thermostabilizing mutations and inhibition of prestin.

2) In Conclusions L218+219: The large distance ($>4 \text{ \AA}$) between Cl⁻ and R399 is ascribed to helix dipoles TM3 and TM10. More quantitative evidence could be obtained from the simulations

- Present a histogram of the Cl⁻ - R399 distance over the course of the simulation.

- Calculate the electric potential in the binding site from the MD simulation (e.g., use the PME electrostatics plugin in VMD) with and without R399 included (set the charge to 0 in the PSF file) and demonstrate that it is due to the helix dipoles.

3) It is not clear how robust the result is that Cl⁻ remains in the binding site (especially as the simulation is rather short with 100 ns) and no spontaneous binding/unbinding was shown.

Show Cl⁻ distance data for both protomers and run another two independent repeats of the simulations to demonstrate reproducibility.

4) Methods: citations are missing for packages such as NAMD, VMD, AMBER, PROPKA; forcefields (CHARMM27, CHARMM36, TIP3P-CHarmm, Amber ff14SB, lipid14, TIP3P), and algorithms (PME, thermostat and barostat, SETTLE, RATTLE). Add missing citations.

5) How stable was the protein (show the RMSD relative to the starting model during the 50 ns equilibration and the 100 ns production simulation)?

6) State clearly if the Cl⁻ ion was placed in the putative binding site or if it bound spontaneously.

Specific comments by Reviewer #1:

This manuscript is the fourth in a burst of papers describing cryo-EM structures of SCL26A5, or prestin, the piezoelectric-like actuator protein of auditory outer hair cells that drives cochlear amplification in mammals. Experimental structures of this protein have been awaited eagerly, to provide the basis for understanding the seemingly unique electromechanical behavior.

The structures presented are technically sound and contribute to a structural framework for further analysis of molecular mechanisms. However, given the mentioned parallel reports, the message is largely confirmatory (maybe with the exception of a sulfate bound structure). The same is true for the larger part of the accompanying functional results reported here, that to a substantial part replicate or extend previously published data.

The approach taken by Futamata et al, differs from the 3 competitive papers in that they determine the structure of a mutationally modified ('thermostabilized') version of prestin, apparently helping to obtain a well-resolved cryo-EM structure. In contrast, the competitors achieve comparable data quality with native prestin isoforms.

Unfortunately, this approach entails the major limitation of the present study as this construct lacks function, i.e. the non-linear charge movement (NLC) thought to reflect the molecular transitions underlying electromotility. Consequently, the obtained structures are highly unlikely to reveal the dynamic behavior of prestin. In fact, the presented structures bound to different anions, which based on functional data are known to drive changes in state occupancy, show the same 'frozen' conformation. In contrast, structures with chloride, sulfate and salicylate obtained by Ge et al (2021, Cell) and Bavi et al. (bioRxiv, 2021) differ from each other and reveal dynamic behavior of prestin. Nevertheless, given the near-synchronous emergence of these data with the mentioned companion papers, as well as the importance of these structures, this report is certainly worth publishing.

We appreciate Reviewer #1's favorable comments on our manuscript.

However two major aspects need to be changed :

1. The authors suggest throughout the manuscript that their data show an elevator-like/rigid-body movement of prestin. However, their data do not provide any additional evidence to such a mechanism, as the structures are essentially identical ('frozen construct') and the functional (mutational) experiments that probe the interface are consistent with dynamics between core and gate domains but cannot specifically show an elevator-like movement. Clearly, the manuscript's title 'reveal the electromotility mechanism' is not supported by the data. Therefore this claim needs to be toned down massively, including the summary scheme in Fig.5. (This said, such a mechanism appears quite likely, particularly in the light of some of the competitors' data, but the results in this manuscript provide no substantial evidence beyond what is already known.)

We agree that the claim should be toned down. We changed the title to “Cryo-EM structures of thermostabilized prestin provide mechanistic insights underlying outer hair cell electromotility”.

2. A comparison to the abovementioned published structures in Ge et al. (*Cell*, 2021), Bavi et al. (*bioRxiv*, 2021), and Butan et al. (*bioRxiv*, 2021) is warranted, given that those data and conclusions are available at this time.

The overall structures of Pres^{TS} are similar to those recently reported by other groups; however, we found notable differences in the core domain position especially in the Cl⁻-bound state (Figs. 4b-c, Supplementary Fig. 16b), in which the core domain is closer to the inward-facing state in Pres^{TS}, whereas it is closer to the outward-facing state in wild-type hPres (Fig. 4d). The new Fig. 4 and Supplementary Fig. 16 in our revised manuscript visually summarize the differences, which are mentioned in the revised manuscript.

Specific points:

3. Logic underlying the generation of ‘thermostabilized’ prestin (l.79 and Suppl. Fig. 2). ‘we engineered the thermostabilized hPres (hPres^{TS}) by replacing multiple amino acid residues with those evolutionally conserved among prestin orthologs. According to alignment in Fig. S2 there are many replacements that - contradicting this statement - do not introduce conserved amino acids, but residues not found in SLC26A5 orthologs. What is the logic behind this?’

In the previous figure, we showed only representative animal orthologs. We performed an HMM search and obtained 10,718 sequences, including eukaryotic SLC26 transporters and allowing duplicated and/or partial sequences (Supplementary Data 1). Using the hPres sequence as the template, the amino acids conserved in more than a certain percentage of species were left in the original hPres sequence, and less-conserved amino acids were replaced with those best conserved in other species, but the residues near the anion binding site were unchanged. Several conservation thresholds from 60 to 95% were set, and the construct with “60% conserved sequence” showed the best thermostability and allowed the structural determination. These points are described in detail in the Methods section of our revised manuscript. We also included a table summarizing the introduced thermostabilizing (TS) mutations (Supplementary Data. 1), and sequence alignments for several eukaryotic prestin homologues (Supplementary Fig.2).

Previous studies have confirmed that this method improves the thermostability of immunoglobulin domains (Steipe et al., *J. Mol. Biol.*, 1994, doi: 10.1006/JMBI.1994.1434), GroEL mini-chaperones (Wang et al., *Protein Sci.*, 1999, doi: 10.1110/PS.8.10.2186), EAAT-1 (Cirri et al., *eLife*, 2018, doi: 10.7554/ELIFE.40110) and xCT transporter (Oda et al., *Protein Sci.*, 2020, doi: 10.1002/PRO.3966). The consensus mutagenesis approach generally increases the

stability of protein folding, by increasing the inter-molecular interactions that are evolutionarily conserved. Accordingly, the introduced mutations are widely distributed throughout Pres^{TS}, and many of them seem to contribute to the protein folding. To show the residues that are conserved/mutated from hPres, we included a new figure with the mutated residues mapped on the Pres^{TS} structure (Supplementary Fig. 3b).

4. L 110.f. *With respect to the identity of the anion binding site only references to non-SLC26 transporters are made. In fact, bindings sites for several SLC26 transporters have been structurally determined (SLC26Dg: Geertsma et al.2015; SLC26A9: Walter et al., 2019; Sultr: Wang et al., 2021). Functionally, anion binding in SLC26A9 (Walter et al., 2019) and even in prestin itself (Gorbunov et al., 2014) have been analyzed in detail.*

Thank you for the comment. We cited these preceding studies in our revised manuscript.

5. L. 115 ff., *coordination of chloride should be compared to results from GE et al., where R399 seems to be not directly involved.*

As suggested, we compared the distances of a bound Cl⁻ to R399 in hPres (Ge et al., Cell, 2021, doi: 10.1016/J.CELL.2021.07.034) vs. Pres^{TS} (this study). The Cl⁻ ion is about 7.1 Å distant from R399 in the hPres structure (Supplementary Fig. 14a), which is longer than the corresponding distance in our present Pres^{TS} structure (4.4 Å) (Fig. 2a). Since Cl⁻-bound prestin is expected to adopt distinct conformational states depending on its external environment (voltage, membrane composition, etc.), it is conceivable that the distance between the bound Cl⁻ and Arg399 differs among different conformational states. Nevertheless, it is also possible that the shorter distance seen in our Pres^{TS} could be ascribed to the thermostabilizing mutations. Since we cannot distinguish these possibilities, we simply described this difference in our paper. Please note that we performed MD simulations for longer periods of time to assess the electrostatic interactions and their contributions to Cl⁻ binding in Pres^{TS}.

6. L. 122.; *S396D/E : mutants were already reported and characterized elsewhere*

We are aware of those preceding studies (references #28 and #29 in our revised manuscript).

7. L.129, Fig. S7: *I am not convinced by this method for measuring membrane targeting, although elegant in principle. It can account for the relative percentage of the protein at the PM, but not the overall expression level. In fact, the examples shown in SF7 suggest that the levels may differ substantially which may contribute to effects seen in NLC recordings. Moreover, there is reportedly a strong variability in membrane localization between individual cells, is this the case here? NLC recordings were done 1 day after induction, but PM localization assay after 2 days. Therefore, I would*

like to see high-quality images of the high PM localization of prestin required for reliably measuring NLC.

Fluorescence cell images are included in our revised manuscript (Supplementary Fig. 8a). These images allow visual assessments of the overall protein expression and plasma membrane targeting of our mTq2-tagged prestin constructs, but only qualitatively. We agree with this Reviewer that our Cy3-based PM targeting assay does not report the overall expression levels. To better address this Reviewer's concern, we measured mTq2 fluorescence (F_{mTq2}) and the optical densities of cells at 660 nm ($\text{OD}_{660\text{nm}}$) on day 1 and day 2 after doxycycline application, which allowed objective and quantitative comparisons of the overall protein expression (using a metric, $F_{\text{mTq2}}/\text{OD}_{660\text{nm}}$). This additional result is included in our revised manuscript (Supplementary Fig. 8b).

8. MD simulations: the position of chloride appears different in the simulation compared to the experimental structure. A quantitative comparison is required. Also, only a single 100ns run of the MD simulation is shown. Were repetitions made?

In revision, we performed 10 independent MD simulations for 1 μs each (10 μs in total) and analyzed the trajectories in detail to gain insight into the Cl^- ion binding, in terms of molecular structure and energetic. Please see the response to Reviewer #4 below for details. We found that the Cl^- ion remained stably within the binding pocket for 1 μs in 5 trajectories, and underwent release from the binding pocket and spontaneous rebinding to the pocket from the bulk solvent in 3 trajectories, showing the stable binding of the Cl^- ion to the binding pocket (Supplementary Fig. 9). The position of the Cl^- ion in the MD simulation is also in line with that in the cryo-EM structure, in which a longer coordination distance between the Cl^- ion and the sidechain of Arg399 was observed (Supplementary Fig. 9). The analysis of the electrostatic interaction energy associated with the Cl^- binding indicated that the interaction with the helix-dipole moments of TM3 and TM10 is responsible for the long coordination distance (Supplementary Fig. 11). These results support our notion and are included in the revised manuscript.

9. Fig 3c: salicylate effect on overall structure: it is difficult to understand which conformation is the salicylate-bound one (this is also true for the movie), but it seems that it is the one with the core domain slightly shifted outwards? The reorientation needs to be quantified. Even considering the limited informational value of the dysfunctional TS-variant, the observation should suggest that salicylate poises the proteins towards an outward-open or intermediate state rather than an inward open, as proposed here, which would be consistent with the data of Ge et al.. Please clarify.

This is an important point. We compared the structures of our thermostabilized mutant (renamed Pres^{TS}) and those of wild-type human prestin (hPres) (Ge et al., *Cell*, 2021, doi: 10.1016/J.CELL.2021.07.034) in detail and found that, in the Cl^- -bound state, Pres^{TS} adopts an

inward-open-like conformation, whereas wild-type hPres adopts a conformation near the outward-open state (Fig.4, Supplementary Fig. 16b). However, the salicylate-bound Pres^{TS} and hPres structures look very similar. These points are discussed in the revised manuscript.

10. *Lengthening of OHCs with salicylate: L148 and Suppl. Movie: time course and reversibility need to be shown to make sure that the effect is via prestin and not by disrupting cellular structures as also described previously. Quantification of results (elongation, statistics) is also required.*

Previous studies showed that salicylate increases C_{lin} of OHCs (Homma & Dallos, *J. Biol. Chem.*, 2011, doi: 10.1074/jbc.M110.185694; Santos-Sacchi et al., *J. Neurosci.*, 2006, doi: 10.1523/JNEUROSCI.4548-05.2006), and that this effect is reversible (Kakehata & Santos-Sacchi, *J. Neurosci.*, 1996, doi: 10.1523/jneurosci.16-16-04881.1996; Tunstall et al., *J. Physiol.*, 1995, doi: 10.1113/jphysiol.1995.sp020765). It was also shown that salicylate does not affect C_{lin} of OHCs lacking prestin (Homma and Dallos, *J. Biol. Chem.*, 2011, doi: 10.1074/jbc.M110.185694, their Fig. S2), indicating the prestin-dependence of salicylate's effect. These previous studies support the interpretation that the observed OHC elongation (Supplementary Fig. 12, formerly Supplementary Fig. 9 in the previous manuscript; and Supplementary Movie 2) is driven by the conformational change of prestin induced by salicylate binding. We confirmed that salicylate-induced OHC elongation is indeed reversible (Supplementary Movie 3). Because the cell length differs among OHCs, the elongation of OHCs by salicylate and their shortening after its removal were corrected for the cell length (before salicylate or before wash). These values, $3.2 \pm 0.7\%$ (elongation, mean \pm SD, n=11) and $-2.9 \pm 1.5\%$ (shortening, mean \pm SD, n=12), have been added in the figure legends of Supplementary Movies 2 and 3 in our revised manuscript. The total filming time of Supplementary Movies 2 and 3 are 116 and 200 seconds, respectively. This information is also included in the figure legends of the Supplementary Movies.

11. *L.177: mutational analysis: results are consistent with the importance of the gate-core interface in prestin function, but certainly do not provide evidence for an elevator-like motion.*

Agreed. Recent prestin structures have shown that the elevator-like motion of the two domains is associated with the NLC generation. Therefore, in the revised manuscript, we follow their discussion and changed the sentence to "*These results suggest that the hydrophobic interactions, mediated by these residues, are also important to facilitate the presumed elevator-like motion and thus for NLC generation*".

Minor points.

12. *Fig. 3d-f: helix labeled 'TM10' should rather be TM1 (with Q97), with TM10 located below?*

We revised Fig.3d-f (Fig. 5a-c in the revised manuscript) and corrected the label.

13. Abstract , l.27 'resulting in distinct modulations of nonlinear capacitance (NLC), playing an important role in electromotility.' Sentence is largely incomprehensible; also, falsely suggests that these are results from the presented study.

We revised the abstract and changed the sentence to “The central positively-charged cavity allows flexible binding of various anion species, which likely accounts for the known distinct modulations of nonlinear capacitance (NLC) by different anions.”

Specific comments by Reviewer #2:

Futamata et al. present the cryo-EM structures of a thermostabilized version of prestin, the cochlear amplifier, with bound chloride, sulfate and salicylate anions. Mechanistic insights into prestin function are provided. The contribution is important, but suffers from the very recent publication by Ge et al. (2021), Cell from the Gouaux group.

We appreciate Reviewer #2's favorable comments on our manuscript. The followings are our point-by-point responses.

Major points:

- Lines 269-270: Sentence “About 40% of the amino acid residues were altered in hPres^{TS} as compared to wild-type hPres (Supplementary Fig. 2).”

Question of principle: Can a protein be named “hPres” having only about 60% identity to human prestin? The reviewer suggest to rename the protein to “Pres^{TS}” not “hPres^{TS}”, because of the important difference in identity.

We renamed our thermostabilized construct “Pres^{TS}” in the revised manuscript, as suggested.

- Please indicate in a Table or list (in Supplementary Information), which mutations were introduced into human prestin to generate hPres^{TS}: From Supplementary Fig. 2 it is difficult and time-consuming to identify the mutations. This will help the reader to see the difference between wt and TS at a glance.

We added a table summarizing the introduced mutations (Supplementary Data. 1) and a figure mapping mutations on the Pres^{TS} structure (Supplementary Fig. 3b). The introduced mutations are distributed throughout the molecule, and not concentrated in specific regions. Most importantly, the residues close to the anion binding site remained unchanged.

Supplementary Fig. 3b The TS mutations mapped on the cryo-EM structure of Pres^{TS}. Residues mutated to those with different or similar properties are colored red and purple, respectively. The non-mutated amino acids are colored dark green.

- Please indicate in the sequence comparison of Supplementary Fig. 2 the degree of similarity and identity, e.g., by introducing the symbols *, . and : below the last sequence (currently hPresTS): you will get this information when performing, e.g., alignment with ClustalW. Again, this is helpful for the reader.

Our new Supplementary Fig. 2 includes symbols to indicate the degrees of conservation.

- Lines 124-126: It is unclear to the reviewer why these Ser residues were mutated. Please add more information about the rationale of this experiment. E.g. what are the distances between S116 and S133 to R399? (please include this information): from the Figure it looks like the Ser residues and guanidinium group of R399 are not in distances allowing interactions. Please also elaborate on the discussion of these mutagenesis results –currently unsatisfactory.

Both S116 and S133 are located opposite the anion binding pocket (behind the positively charged R399). S133 is proximal to R399, and the oxygen atom of S133's hydroxyl group and the NE atom of R399 are located nearby, within 3.7 Å. S116 is rather apart from R399 by about 7 Å, but when mutated to Asp/Glu, it could electrostatically interact with R399, either indirectly through Gln94 or directly if the rotamer of R399 changes. Importantly, in contrast to S396D, these mutants (S116D/E and S133D/E) abolished NLC, suggesting that the proper charge balance upon anion binding is critical for NLC generation. The following sentences were added in our revised manuscript, for clarity:

“We also examined the roles of S116 and S133, which are located near Arg399 but on the opposite side from the anion binding pocket. S133 is proximal to R399, while S116 is rather apart but connected to R399 through Gln94. These residues are likely to form electrostatic interactions with Arg399. Mutation of these residues to Glu or Asp (i.e., S116D, S116E, S133D, and S133E) abolished NLC (Fig. 2c).”

- MD simulation experiment in Supplementary Figure 8: Please provide and display at least one additional replica to show reproducibility of these results.

We performed 10 independent MD simulations for 1 μ s each (10 μ s in total) and analyzed the trajectories in detail to gain insight into the Cl^- ion binding, in terms of molecular structure and energetics. Please see the response to Reviewer #4 below for details. We found that the Cl^- ion remained stably within the binding pocket for 1 μ s in 5 trajectories, and it underwent release from the binding pocket and spontaneous rebinding to the pocket from the bulk solvent in 3 trajectories, showing the stable binding of the Cl^- ion to the binding pocket (Supplementary Fig. 9). These results were updated, and the discussion was also slightly modified in the revised manuscript.

- Figure 2a and b: it seems that the rotamers of S396 are different in the chloride and sulfate bound structures. Please discuss this, e.g., are there new interactions? Please also introduce distance, it is difficult to estimate distances in Figures (general issue). For example, is in Fig. 2b S396 O atom in hydrogen bond-distance to R399 N atoms?

The resolution of the SO_4^{2-} -bound Pres^{TS} structure (3.52Å) is not high enough to unambiguously determine the rotamer of a small residue such as serine. Since the cryo-EM density of SO_4^{2-} -bound Pres^{TS} at S396 can reasonably accommodate the rotamer of S396 found in the Cl^- -bound Pres^{TS} structure, we adopted it in our updated SO_4^{2-} -bound Pres^{TS} structure (Fig. 2), to not inadvertently mislead readers. The distance information is also included in our updated Fig. 2.

- Figure 2b: Please show the environment around the sulfate ion in a panel or Suppl. Fig. What are the interactions of the protein with the sulfate anion? Please show and discuss.
Sulfate

The SO_4^{2-} density was relatively low, likely due to its low binding affinity to prestin, but the observed cryo-EM density fit a SO_4^{2-} ion and seemed to be coordinated by the F397 amide and Q454 near the central anion binding pocket. Its binding is probably competitive with Cl^- , but Cl^- is still presents at as much as 300 mM even in the SO_4^{2-} conditions, and the cryo-EM map shows the mixture of the Cl^- - and SO_4^{2-} -bound states. This new information was added in our updated Fig. 2 and the main text was revised accordingly.

- Paragraph on 'Inhibition by salicylate': What is the binding constant of salicylate to prestin? Any numbers from the literature? Please comment on this, readers will want to have an impression on the affinity of this inhibitor.

The binding affinity of salicylate to prestin embedded in the cell membrane was estimated to be about 20 μ M (Oliver et al., Science, 2001, doi: 10.1126/science.1060939) However, the salicylate binding affinity in a detergent-solubilized condition has not been reported. Thus, we performed isothermal titration calorimetry (ITC) experiments using prestin in detergents, and determined the salicylate binding affinities (K_D) to be 7.08 mM and 12.1 mM for hPres and Pres^{TS}, respectively

(Supplementary Fig. 13), suggesting that the TS mutations did not severely affect salicylate binding and that over 70% of the anion binding site was occupied by salicylate under our experimental conditions. This new information is included in our revised manuscript.

- Lines 161-163, statement: “We found that S398A also makes prestin insensitive to salicylate, further affirming the importance of the small polar residue for salicylate binding (Fig. 3b, Supplementary Fig. 9b).” What about S396A? – on Fig. 3a also this residue interacts with the carboxyl group of salicylate. Please add functional data or reference to literature on the effect of -OH group removal. Please indicate distances in Fig. 3a (S396 and S398 to salicylate).

We found that S396A abolished NLC (Fig. 2c) and thus we could not determine the effects of this mutation on salicylate binding. The distance information is now included in our revised Fig. 2.

- Lines 165-166: Rigid body displacement between the core and gate domains: Please quantify this displacement, e.g., indicating distances between the two states, moving angle of a specific transmembrane helix, etc. From this statement the reviewer cannot assess if the displacement is relevant or not.

The difference in the displacement of the core domain (the TM3/TM10 region) with respect to the gate domain (using S398 in TM10 as a reference point) is about 2.8 Å. This information is included in Supplementary Fig. 16a (former Supplementary Fig. 15) in our revised manuscript. We also summarized the relative locations of TM3/TM10 with respect to the gate domain found in the prestin structures reported herein and other recent studies (Supplementary Fig. 16b).

- Line 170: “Gln97 and Asn447 partly constitute the chloride binding site, and their mutations...”. Please show and mention also these residues in Figure 2, where the chloride binding is shown and discussed.

We included Q97 in Fig. 2 N447 is located near the Cl⁻ binding site. However, since it seems unlikely that it directly contributes to Cl⁻ binding (please see Fig. 5), we decided not to show N447 in the figure for clarity. Accordingly, the wording “**constitute** the chloride binding site” was changed to “**located near** the Cl⁻ binding site”

- Last paragraph of “Inhibition by salicylate”. If a negative charge is mutated into a positive (E293K), and hydrophobic residues into polar/charged ones (V353D, V353E and V353N), the reviewer is not astonished to see that the function is reduced or abolished. Did the authors perform less radical mutations, e.g., E293A or E293Q, or V353F, V353Y or V353W? Please introduce results into manuscript.

Thank you for the comment. A previous study reported that E293Q abolished NLC (Bai et al.,

Biophys. J., 2009, doi: 10.1016/j.bpj.2008.12.3948). We measured NLC for E293A, V353F, V353Y, and V353W. Briefly, E293A greatly reduced NLC, but did not abolish it. Regarding V353, substitutions to other hydrophobic residues with similar or smaller sizes only mildly affected NLC (V353A, V353I, and V353L), whereas substitutions to larger hydrophobic residues greatly decreased or abolished NLC (V353F, V353Y, and V353W). These results further support our claim that the electrostatic and hydrophobic interactions at the core/gate domain interface are important for the elevator-like motion. These results are included in our revised manuscript (Fig. 5).

- Paragraph: 'Dimerization of prestin' (page 5). V to G, and I to G are (again) radical mutations, when considering the high flexibility introduced by glycine and the fact that this amino acid is not chiral. Said differently, I am not astonished about the dramatic effect of such mutations on the function. What about the V499A and I500A mutations, i.e., how is the effect on the NLC ?

A previous study (Homma et al., *J. Biol. Chem.*, 2013, doi: 10.1074/jbc.M112.411579) found that amino acid substitutions at the V499 site variously affected the NLC of prestin (please see the figure summarizing their findings below). Note that V499A significantly reduced both the peak magnitude of NLC and its voltage sensitivity (broadened NLC). Interestingly, we found that I500A did not affect NLC much, while I500L significantly reduced the peak magnitude of NLC and its voltage sensitivity, similarly to V499A. Collectively, these observations support the importance of the V499/I500 site for the normal function of prestin. These new results are included in Fig. 6b in our revised manuscript.

Homma et al., *J. Biol. Chem.*, 2013, doi: 10.1074/jbc.M112.411579 (their Fig. 4)

- Discussion: 4.49 Å for the distance between Arg399 and the chloride ion is relatively large, and uncommon. Did the authors consider the possibility to have water bridging Arg399 and chloride ion? What is the local resolution in this area (area around Arg399 and chloride ion)?

The local resolution is better in the TM domain, and about 3.3 Å around the anion pocket, where the R399 side chain is clearly visualized. As the Reviewer pointed out, the distance between Cl⁻ and R399 is relatively large, but this is probably because the Cl⁻ ion is accommodated between

the dipole moments of TM3 and TM10 and the R399 side chain. In wild-type hPres, Cl⁻ and R399 are over 7 Å apart. We added the comparison of the anion pockets (Supplementary Fig. 14).

- The density maps are not well documented. Please add in Suppl. Information for all structures: i) Local resolution (2 views, e.g., top and side view) calculated using MonoRes or another program and ii) Euler angle distribution (again two views).

We added the maps with local resolution and the angular distributions for the three structures in Supplementary Figs. 4-6.

- The rotamer outliers of 7V75 (salicylate) are with 5.7% high (also those from the other two structures are relatively high). Are these outliers in functionally important regions? Please comment.

Those residues with outlier rotamers are located on the loops of the STAS domain or in the extramembrane region, where the densities are not clearly visualized. As in the response below, we did not apply strong restraints on the rotamer, because doing so may overestimate the quality of the maps.

- The RMSD from the bond angles are quite high. Together with the high % of rotamer outliers, do the authors believe that there is room for improving the model?

Typical standard deviations of bond angles range from 1.5 to 3.0 degrees, and the RMSD bond angle of ~1.8 degrees would not be considered as high. Actually, as in the PDB validation reports, the bond angle RMSD values are around 1.0. The rotamer outliers are relatively high, if compared with those refined with rotamer restraints, which we did not use. There should be room for improvement, but we believe our models have the sufficient quality that can be achieved at these resolutions.

- Please provide an SDS-PAGE gel analysis (e.g., in Suppl. Fig. 1) of the protein used for SEC and cryo-EM study. Currently not documented.

We included the SDS-PAGE gel of the purified protein that was used for cryo-EM analysis in Supplementary Fig. 1b, e.

- Please expand the discussion, on how the benzene ring should sterically restrict the elevator-like motion of the core/gate domains: from Fig. 3f, 5c, this is difficult to imagine.

We included a new Supplementary Fig. 14 to show the detailed interactions of the bound salicylate within the hPres and dPres structures in addition to our salicylate-bound Pres^{TS} in Fig. 3. While the carboxylate anionic moiety of salicylate is attracted toward the central positive-charged and hydrophilic pocket, the bulky benzene ring is oriented toward the hydrophobic residues on the

gate domain. In all structures, the benzene ring fit into the hydrophobic pocket constituted by A218, I (V) 221, V353, L448, and L (M) 451 (residues in parentheses indicate those of hPres and dPres) (Fig. 3c, Supplementary Fig. 14g, i). Thus, salicylate may function as a “wedge” to fix the two domains in a conformation close to the inward-open state (Fig 3d, Supplementary Fig. 14f-i), and it may also sterically restrict the structural change toward the outward-facing-like state similar to the Cl⁻-bound wild-type hPres (PDB: 7LGU, Ge et al., Cell, 2021, doi: 10.1016/J.CELL.2021.07.034) and dPres (PDB: 7S9E, Bavi et al., Nature, 2021, doi: 10.1038/s41586-021-04152-4). The structural comparison and discussion are augmented in the revised manuscript.

| - Please add a summarizing last paragraph (in Discussion) or a Conclusion.

Thank you for this suggestion. We added a concluding paragraph in the revised manuscript to summarize our findings.

| - Please provide in Suppl. Information the codon-optimized DNA sequence of hPres^{TS}.

We added the codon-optimized DNA sequence of Pres^{TS} in Supplementary Data 2.

| - Subchapter: “Purification of hPres and hPres^{TS}”: Please consider that you do not only explain purification here, but also overexpression: Please check title, currently suboptimal.

We changed the subheading to “Expression and purification of hPres and Pres^{TS}”

| - You used 10 mM sulfate and 30 mM salicylate for cryo-EM experiments. What are the binding constants of these two anions for hPres and your hPres^{TS} (see also my previous question above about K_d of salicylate)? How many fold excess of ligand (anion) was finally added for grid preparation?

As in the response to the above comment, the K_d of salicylate to Pres^{TS} was estimated to be about 20 μM in cells, but was 12.1 mM in our ITC experiments, using purified protein samples solubilized in detergent. The concentration (30 mM) used in the cryo-EM analysis would thus allow salicylate binding to over 70% of Pres^{TS}. Since salicylate-free Pres^{TS} likely adopts a distinct conformation, they should be classified into minor classes (so-called “garbage classes”) during the 2D- and 3D-classifications. Cryo-EM analysis often yields only a major class of conformations when the population of other conformations is small. Therefore, salicylate-free Pres^{TS} is unlikely to have compromised our effort to define the salicylate-bound Pres^{TS} structure.

Regarding SO₄²⁻, we also performed ITC experiments, but the measurement suffered from large noises probably due to SO₄²⁻'s dissolution heat, and consequently, we could not determine the K_d value for SO₄²⁻ binding. In the cryo-EM analysis, we also tried solutions containing SO₄²⁻ as high

as 200 mM SO_4^{2-} ions, but they produced “bad-ice” on the grids and did not allow structural determination. Due to this technical issue, we ended up solving the structure of SO_4^{2-} -bound Pres^{TS} using a solution containing 10 mM SO_4^{2-} and 300 mM Cl^- . As anticipated, a weak density at the Cl^- site (found in Cl^- -bound Pres^{TS} structure) remained, but an additional density appeared near the central anion pocket. These results suggest that SO_4^{2-} could bind to Pres^{TS} under our experimental conditions. We slightly toned down the discussion as follows: “*despite the presence of Cl^- ions at the same concentration (300 mM), a density instead appears nearby, where the solvent is more accessible and the F397 amide and Q454 side-chain could coordinate an ion (Fig. 2a and 2b). We supposed that SO_4^{2-} competes with Cl^- but is accommodated in the anion binding site in a slightly different manner, and thus modeled a sulfate ion onto this density. It is conceivable that the known dependence of NLC on various anion species can be partly ascribed to subtle differences in the binding positions among anions.*”

— *Important: Considering the very recently published structures by Ge et al. (Sept. 2, 2021) Cell from the Gouaux group, please perform a comparison between the structures of this work and Ge et al. (i.e., Pres-Cl, Pres-Sal and Pres-Sulfate). Please incorporate the comparison in manuscript and discuss.*

We appreciate this comment. Fig. 4 and Supplementary Fig. 16b visually summarized the differences between our Pres^{TS} structures and wild-type prestin from human (Ge et al., Cell, 2021, doi: 10.1016/J.CELL.2021.07.034), dolphin (Bavi et al., Nature, 2021, doi: 10.1038/s41586-021-04152-4), and gerbil (Butan et al., Nat. Commun., 2022, doi: 10.1101/2021.08.03.454998). It seems that thermostabilizing mutations stabilize Pres^{TS} in the inward-open-like conformation in the Cl^- -bound state, while salicylate stabilizes both hPres and Pres^{TS} in a similar conformation. This is discussed in the second paragraph of the “*Inhibition by salicylate*” section in the revised manuscript.

Minor points:

— *Introduce the abbreviation STAS for non-expert readers*

STAS stands for “Sulfate Transporter and Anti-Sigma factor antagonist”. We added this unabbreviated term in the revised manuscript.

— *Please check Supplementary Fig. 6: there are fine horizontal and vertical gray lines, which in my opinion do not have to be there (probably involuntarily introduced during figure assembly).*

Thank you for spotting the issue. We fixed it.

— *I guess it is more appropriate to write “SLC26” and not “SLC26A”, i.e., only “SLC26” if not a specific transporter is addressed, but stated in a general context. Please check.*

We use “SLC26” throughout the revised manuscript.

- Please use one style only, i.e., do not mix one letter and three letter amino acid coding: currently mixed in text and Figures (e.g., also Suppl. Fig. 8a three-letter code).

Thank you for pointing this out. We decided to use the one letter code throughout the manuscript.

- Do not mix uncapitalized and capitalized panel labeling, e.g., compare line 222 and line 227.

We decided to use lower cases for all figure panels in our revised manuscript.

Reviewer #3 (Remarks to the Author):

Reviewer comments for «Cryo-EM structures reveal the electromotility mechanism of prestin, the cochlear amplifier.

In this study a thermostabilized Prestin variant is studied by cryo-EM and corresponding functional assays. The authors present structures of Prestin bound to chloride, sulfate and salicylate. They identify residues important for binding of chloride and sulfate, and for inhibition by salicylate. The study is nicely carried out, well written and interesting. Their EM data is convincing, and conclusions and speculations made from the structures seems reasonable and are well supported by their NLC data. I still however have some comments that I think should be addressed.

We appreciate Reviewer #3's favorable comments on our manuscript. Followings are our point-by-point responses.

1.

Initial efforts to solve the structure of human Prestin (hPres) by cryo-EM was not successful due to problems in obtaining homogenous protein fractions from size exclusion chromatography. In order to obtain a more stable variant of hPres, using a consensus mutagenesis approach they make a hPres thermostabilized variant hPresTS which they use to solve the chloride, sulfate and salicylate bound structures. In the thermostabilized Prestin variant, approximately 40% of the amino acid residues were altered compared to hPres WT. The thermostabilized Prestin variant has no detectable NLC activity.

When making a thermostabilized Prestin variant, in which 40% of amino acids are mutated to consensus residues, the authors should consider renaming the thermostabilized Prestin variant PresTS instead of hPresTS as the variant is just as much a generic Prestin variant as a human Prestin variant. I also think that the number 40%, or a more precise description of the differences between PresTS and Pres WT should be included in the result section.

Thank you for the suggestion. The construct is now named Pres^{TS} throughout the manuscript. We

added detailed descriptions about how we determined the mutations introduced to hPres to generate five Pres^{TS} constructs, and a table summarizing the sequence conservation among species and the introduced mutations. We also included the DNA sequence of the Pres^{TS} construct used for structural determination in our revised manuscript (Supplementary Information).

It is also stated that partial reversion of the TS mutations neither rescued NLC activity or allowed structural determination, but there is no description of which partially reverted hPres variants that were tested, or how the SEC profile of these variants looked compared to hPresWT and hPresTS. I am also wondering whether the hPres WT had measurable NLC activity or if it was only HgPres that could be used in NLC measurements.

We made five Pres^{TS} constructs. The SEC profiles of these constructs and WT hPres are now provided in Supplementary Fig. 3. All Pres^{TS} mutants were expressed at very high levels in HEK cells (due to stabilization), but none showed clearly detectable NLC. Structural determination was successful only for the Pres^{TS} construct with highest amino acid replacement (40% consensus mutant), as described in the manuscript.

NLC was measurable in cells expressing WT hPres, consistent with a recent report by Ge *et al.*, 2021. However, we experienced technical difficulty in measuring the NLC of hPres, due to its very low expression. To facilitate our efforts to systematically characterize a large number of prestin mutants, we decided to use *Heterocephalus glaber* (naked mole-rat) prestin (HgPres) mainly due to its good expression. The amino acid sequence of prestin is highly conserved among species (96% identical between hPres and HgPres).

2.

Further, they described the Prestin variant as thermostabilized but does not show, or describe any assay showing that the altered Prestin variant is actually more thermostable. From Figure S1 hPresTS elutes in a nice monodisperse peak (contrary to hPres WT), but additional experiments showing thermostabilization should be performed in order to claim that hPresTS actually is thermostabilized.

We performed an FSEC-TS analysis (Hattori *et al.*, *Structure*, 2012, doi: 10.1016/j.str.2012.06.009) for hPres WT and Pres^{TS} and confirmed that the TS mutations (40%) in fact increase the thermostability of the prestin protein (T_m increased by 13.4 °C). This result is included in Supplementary Fig. 3 in our revised manuscript, and is also shown below.

Supplementary Fig. 3e, f. FSEC-TS analyses of hPres (e) and Pres^{TS} (f).

3.

Recently 4 Cryo-EM structures (2.3 – 4.3 Å) of Cl⁻ bound and salicylate bound hPrestin was published (Ge et al, Cell, 2021) but there is no reference to these structures or this paper in the manuscript. These Prestin structures have been available online since August 2021 and should be referred to in the introduction.

Further, as these are structures of WT hPrestin, the current manuscript could benefit from a discussion around similarity or differences between the structures of hPresWT and hPresTS.

We compared our Pres^{TS} structures to the hPres structures reported by Ge et al. Briefly, the Cl⁻-bound states are notably different between hPres vs. Pres^{TS}, whereas the salicylate-bound states look very similar to each other. These results (Fig. 4 and Supplementary Fig. 16) and discussion are included in the revised manuscript.

4.

In the description of the anion binding site Arg399 is pointed out as a key residue in coordination of the chloride ion, but it is not among the residues that are mutated and tested in NLC measurements. Although the relevance of Arg399 in the coordination of Cl seems very likely it would be nice to either include data showing this or cite earlier work describing the importance of this residue.

R399 is reportedly essential for the expression of NLC (Bai et al., *Biophys. J.*, 2009, doi: 10.1016/j.bpj.2008.12.3948; Gorbunov et al., *Nat. Commun.*, 2014, doi: 10.1038/ncomms4622). We added the sentence “R399 is a highly conserved residue within the SLC26 family, and its critical roles in anion binding and NLC have been extensively studied” and cited these two previous reports in our revised manuscript.

5.

In the section describing dimerization of Prestin, a figure showing more of the dimerization interface, including also the hydrophilic residues mentioned in the discussion should be included. Are these

| residues for “stable domain swapped dimer formation” and NLC generation?

The C-terminal cytosolic domain of prestin seems to contribute to dimerization. However, since we do not provide experimental evidence supporting (or opposing) such a possibility, we decided not to describe or discuss this in our revised manuscript. The readers can entertain any possibilities using the PDB data of prestin structures provided by us and by others.

| Further the relevance of lipid acyl chains and cholesterols (also shown in fig 4C and 4D) on Prestin stabilization and the effects of lipid manipulation on the function of the protein should be explained more thoroughly. To me, these final statements in the result section is a bit cryptic.

Modulation of prestin activity by lipids has long been recognized and thoroughly established by multiple previous studies; however, its underlying mechanism remains undefined. Thus, finding lipid molecules on the prestin structure is of great significance, as it provides mechanistic insights into prestin function modulation by lipids. We cited key studies highlighting the importance of lipids for prestin function and added the following sentences at the end of the Results section to clarify the importance of this finding: “Prestin activity (electromotility and concurrent NLC) is sensitively modulated by the lipid composition and thickness of the cell membrane. For example, the voltage operating point of prestin is tuned by the cholesterol contents. Therefore, the lipid molecules observed on the TMD surface and at the dimer interface provide significant insights and suggest their direct involvement and contribution to the molecular conformational changes of prestin.”

| 7.

| Finally I have a few small comments on the methods section:

| Line 266: a citation to the used pEG BacMam vector or a description about what modifications that were done to the vector should be included.

We used the original pEG-BacMam vector for the C-terminal GFP fusion.

| Line 301: Could you include the concentration or relative amount of TEV that was used?

Approximately 1 mg of TEV protease was added to a 10 mg dose of GFP-tagged Pres^{TS}. We added this information in the Methods.

| Line 347 and 352: From the PDB validation report it seems like parts of the structures could not be modeled. A description of which parts of Prestin that was modelled and which parts that were excluded / could not be modeled due to lack of density in EM data should be included.

The final model includes the amino acid residues of T13–K580 and Y614–S725, while the omitted

loop in the STAS domain is not visible in the cryo-EM maps. We explained this in the Methods.

Line 397: Was there any missing residues in the structure, between residue 58-506, and where these modeled in or left out from MD simulations?

We were able to build a model for this entire region, so there are no missing residues. Residues 58–506 correspond to the TMD, and the MD simulations were performed as the TMD monomer, lacking the STAS domain, to minimize the simulation size. Therefore, we only focused on the anion binding/dissociation and the local conformational changes within the core domain, and have clarified this in the revised Methods.

Reviewer #4 (Remarks to the Author):

This paper presents three cryo-EM structure of human prestin (using a thermostabilized mutant): a Cl⁻ bound structure at 3.52 Å resolution, a sulfate (SO₄²⁻) bound structure (3.52 Å), and a salicylate bound structure (3.57 Å). The thermostabilized protein does not show the NLC (non linear conductance) typical for active prestin so the protein structure is that of a non-functional or inhibited prestin. Functional measurements (NLC) of mutants (of the WT protein) are used together with a short MD simulation to gain insight into the mechanism of prestin.

Overall, the paper falls short in really revealing the electromotility mechanism of prestin (as promised in the title "Cryo-EM structures reveal the electromotility mechanism of prestin, the 2 cochlear amplifier") because the data does not contain evidence of the conformational changes ("elevator like", as described by the authors) that they hypothesize are required for the area changes that underly prestin's electromechanical action in the outer hair cell membrane. The paper presents a number of additional data points, in particular with regard to the potentially important interactions between core and gate domain, that add to the recent papers on prestin, namely the recent cryo-EM prestin structures from human [1], dolphin [2], and gerbil [3].

*[1] J. Ge, J. Elferich, S. Dehghani-Ghahnaviyeh, Z. Zhao, M. Meadows, H. von Gersdorff, E. Tajkhorshid, and E. Gouaux, "Molecular mechanism of prestin electromotive signal amplification," *Cell*, vol. 184, no. 18, pp. 4669–4679.e13, 2021.*

*[2] N. Bavi, M. D. Clark, G. F. Contreras, R. Shen, B. G. Reddy, W. Milewski, and E. Perozo, "Prestin's conformational cycle underlies outer hair cell electromotility," *Nature*, vol. e-pub, 2021.*

*[3] C. Butan, Q. Song, J.-p. Bai, W. Tan, D. S. Navaratnam, and J. Santos-Sacchi, "Single particle cryo-em structure of the outer hair cell motor protein prestin," *bioRxiv*, 2021.*

We appreciate this comment. We slightly toned down our manuscript by changing the title to "Cryo-EM structures of thermostabilized prestin provide mechanistic insights underlying outer hair cell electromotility" and included comparisons of our prestin structures to those reported by

others, in our revised manuscript.

*1) It is always difficult to write a paper when similar work is being published almost at the same time. I would *suggest* to include a comparison to the already published structures and perhaps find an explanation for why the salicylate bound, thermostabilized structure appears to differ from the salicylate bound structures in [1] and [2] (at least as far as I can tell). This might shed some light on the effect of the thermostabilizing mutations and inhibition of prestin.*

Thank you for the kind comment. We fully agree with the Reviewer's suggestion. The structural comparison with the recent prestin structures revealed that Pres^{TS} is likely to be stabilized in an inward-open conformation upon Cl⁻ binding, which is inconsistent with the outward open-like structures found in the Cl⁻-bound prestin structures solved by others (Fig. 4b). These discrepancies are described and discussed in the revised manuscript.

2) In Conclusions L218+219: The large distance (>4 Å) between Cl- and R399 is ascribed to helix dipoles TM3 and TM10. More quantitative evidence could be obtained from the simulations

- Present a histogram of the Cl- - R399 distance over the course of the simulation.*
- Calculate the electric potential in the binding site from the MD simulation (e.g., use the PME electrostatics plugin in VMD) with and without R399 included (set the charge to 0 in the PSF file) and demonstrate that it is due to the helix dipoles.*

3) It is not clear how robust the result is that Cl- remains in the binding site (especially as the simulation is rather short with 100 ns) and no spontaneous binding/unbinding was shown.

Show Cl- distance data for both protomers and run another two independent repeats of the simulations to demonstrate reproducibility.

These are very important points. We thank the reviewer for the comment. In the revision, we performed 10 independent MD simulations for 1 μs each (10 μs in total) from different initial conformations to examine the structures and energetics of the Cl⁻ binding (Supplementary Figs. 9 and 10). We employed a monomer part of the transmembrane domain in the MD simulation, which considerably reduced the computational cost and significantly improved the simulation and analysis statistics (see below). The initial structures were sampled from an MD simulation with distance constraints that kept the bound Cl⁻ ion within the binding site for 500 ns. In the 10 trajectories for 1 μs each without constraints, we found that the Cl⁻ ion remained stably within the binding pocket in 5 trajectories, and it underwent release and spontaneous rebinding to the pocket from the bulk solvent in 3 trajectories, as observed in a previous MD simulation (Ge et al., Cell, 2021, doi: 10.1016/J.CELL.2021.07.034) showing the stable binding of the Cl⁻ ion to the binding pocket with occasional dissociation in the inward-open conformation (Supplementary Fig. 9). The RMSDs of the protein (Supplementary Fig. 10) also showed that the protein

conformations in the Cl⁻ ion bound state remained stable during the 1 μs simulation, while slight increases of the RMSD occurred upon the release of the Cl⁻ ion (e.g. Run3) were observed. The position of the Cl⁻ ion in the MD simulation is also in line with that in the cryo-EM structure where a longer coordination distance between the Cl⁻ ion and the sidechain of R399 was observed (Supplementary Fig. 9).

We also analyzed the contributions of the R399 sidechain and the helix dipole moments of TM3 and TM10 to the Cl⁻ binding, by calculating their electrostatic interactions with the bound Cl⁻ ion for the MD trajectories (Supplementary Fig. 11. See also Methods for details of the calculation). The result revealed that the contribution from the helix dipole moments of TM3 and TM10 is larger than that by R399, explaining the large coordination distance in the cryo-EM structure and the simulations.

We revised the MD simulation part and the discussion in the revised manuscript.

4) *Methods: citations are missing for packages such as NAMD, VMD, AMBER, PROPKA; forcefields (CHARMM27, CHARMM36, TIP3P-CHarmm, Amber.ff14SB, lipid14, TIP3P), and algorithms (PME, thermostat and barostat, SETTLE, RATTLE). Add missing citations.*

Thank you for pointing this out. The corresponding references were added to the revised manuscript.

5) *How stable was the protein (show the RMSD relative to the starting model during the 50 ns equilibration and the 100 ns production simulation)?*

As in the response to the comment 2, we now show the RMSD values during the simulations in Supplementary Fig. 10.

6) *State clearly if the Cl⁻ ion was placed in the putative binding site or if it bound spontaneously.*

As in our response to comment 2, we performed 10 independent MD simulations for 1 μs each (10 μs in total) from different initial conformations. The Cl⁻ ion remained stable in the pocket in 5 of the 10 simulations. The release of the Cl⁻ ion was observed in the other 5 simulations, but its spontaneous rebinding to the pocket from the bulk solvent was also observed in 3 trajectories. These results basically support the relatively stable binding of Cl⁻ to the central pocket, as discussed in the revised manuscript.

REVIEWERS' COMMENTS

Reviewer #1 (Remarks to the Author):

The manuscript was improved substantially and overall my concerns were addressed satisfactorily.

In particular, the logic behind generating PresTS is now much clearer.

In this respect one concern remains: given that the general strategy was to mutate towards residues conserved across eukaryotic SLC26 isoforms (rather than electromotile mammalian SLC26A5 orthologs): how far does PresTS represent a prestin (i.e. mammalian A5) protein? Or is the examined protein rather a prototypic pan-SLC26 protein that reveals more about SLC26 transporters in general rather than specifically about prestin and its electromotile mechanism? I would suggest to provide a meaningful measure of the sequence similarity of PresTS to mammalian prestin and other (mammalian) SLC26 isoforms, such as SLC26A5, A4 and so on.

Minor points:

I. 222 “upon the transition toward the outward-facing state (Cl⁻-bound hPres)”

Wording seems to suggest that the solved Cl⁻-bound hPres structure is outward-facing, while in fact it is occluded.

I. 203 onwards: for readability, better use a new subheading as this section deals with inter-domain interactions rather than salicylate effects

Reviewer #2 (Remarks to the Author):

Dear Authors,

Great work, it was a pleasure reading the revised version and the point-by-point discussion, which was performed very carefully.

Congratulations,
Dimitrios Fotiadis

Reviewer #3 (Remarks to the Author):

I am happy with the clarifications, and changes done to the manuscript.

Reviewer #4 (Remarks to the Author):

The authors performed an additional 10 x 1 μ s of MD simulations that provide sufficient evidence for Cl⁻ binding to the putative binding site. They also performed additional analysis to more clearly define the role of the TM3/TM10 helix dipole moments to stabilize an anion in the binding pocket.

Other important changes:

A detailed discussion of the similarity between PresTS:Cl⁻ and PresTS:sulfate structures was added. [To this reviewer, the discussion in conjunction with the comparison to published structures suggests that the thermostabilization might have shifted (and locked) PresTS in an "down"-like state, although I would not dare to speculate that it resembles the physiological expanded state.]

Additional ITC measurements for the K_d of salicylate binding indicate that the binding site in hPres and PresTS bind salicylate at similar strength.

An interesting comparison between recently published prestin structures and the PresTS results was added.

I have a few remaining minor comments:

1) The One-sentence summary "Cryo-EM reveals the molecular mechanism of prestin." is still overselling the results and should be toned down.

2) The Introduction makes no mention of the 3 other prestin papers---from reading only about other SLC26 transporters one is lead to believe that there's nothing else known about prestin's structure. Please adhere to scholarly standards and cite the other papers and give an accurate picture of the current state of the field.

3) L100 "These observations suggest that the consensus mutations collectively stabilized the structure of PresTS."

I don't follow this conclusion. How is it possible to infer from "loss of function" that PresTS was stabilized? Or maybe the paragraph-breaking is confusing and the concluding sentence was supposed to primarily refer to the SEC and cryo-EM structure determination data?

Note that the authors can make the case that their structure, even if non-functional, provides sufficient insights to generate new hypotheses. For example, they functionally probe residues in HgPres.

4) L127 "probably corresponds to a Cl⁻ ion coordinated by R399 and S396"

With the somewhat large Cl⁻ - R399 distance and the finding of the importance of the helix dipoles, does it still make sense to call R399 a coordinating residue?

5) On L154 "Notably, the coordination distance between the Cl⁻ ion and R399 found in our cryo-EM PresTS structure is 4.49 Å,..." a distance is quoted that does not match with the data in Suppl Fig 9, which presents distances > 6 Å for Cl⁻ - R399.

In Suppl Fig 9, the Cl⁻ - R399 distance is calculated to the C_{zeta} of the arginine. It is not clear if the 4.49 Å were calculated differently. Please clarify.

Presumably, it would be more meaningful to calculate the shortest distance for each simulation timestep between the anion and the positively charged guanidinium nitrogens because coordination distances are typically given for the ligand heavy atom. I suggest the authors add this distance $\min[d(\text{Cl}^-, \text{N}_{\text{eta}1}), d(\text{Cl}^-, \text{N}_{\text{eta}2})]$ to Suppl Fig 9.

6) typos

L161 "ans" -> "and"

L570 "perfoemd" -> "performed"

7) L210-212 "Pres adopts an inward-open-like state, whereas hPres adopts an outward open-like conformation (Fig. 4a-c, d-left, Supplementary Fig.16b)."

Just a comment: In the context of prestin I find it confusing to label the hPres:Cl⁻ structure as "outward-open" as this implies to me that the binding site has switched accessibility from the cytosolic to the extracellular solution. That makes sense for transporters but less so for prestin.

8) L217 "... structural determination of a Cl⁻-bound elongated state of prestin that was not captured in recent studies by others"

There's no evidence presented that the PreTS:Cl- structure is an "elongated state" (probably meant in the sense of "expanded"). There's no quantitative measure of similarity provided with putatively expanded structures or the direct change in area. Either provide evidence and clarify or remove.

9) MD methods

Were really "constraints" used (i.e., the exact distance was held fixed with something like SHAKE) or were these "restraints" (e.g., harmonic pseudo bonds with a specific force constant)? Please clarify.

Were the electrostatics calculations performed with CPPTRAJ, AMBER, or something else, were periodic boundary conditions taken into account, and was the relative electric permittivity set to 1 (vacuum) or another value?

10) Fig 4c and Suppl Fig 16b

The schematic comparison of the different structures is very informative and a good example for clean graphical design. But it's not clear how the authors arrived at the ordering -- please describe in Methods or a caption what quantity was calculated to compare the different structures.

I

Specific comments by Reviewer #1:

The manuscript was improved substantially and overall my concerns were addressed satisfactorily.

We appreciate Reviewer #1's favorable comments on our manuscript.

In particular, the logic behind generating Pres^{TS} is now much clearer.

In this respect one concern remains: given that the general strategy was to mutate towards residues conserved across eukaryotic SLC26 isoforms (rather than electromotile mammalian SLC26A5 orthologs): how far does Pres^{TS} represent a prestin (i.e. mammalian A5) protein? Or is the examined protein rather a prototypic pan-SLC26 protein that reveals more about SLC26 transporters in general rather than specifically about prestin and its electromotile mechanism? I would suggest to provide a meaningful measure of the sequence similarity of Pres^{TS} to mammalian prestin and other (mammalian) SLC26 isoforms, such as SLC26A5, A4 and so on.

Using FastTree, we generated a molecular phylogenetic tree based on the primary amino acid sequence alignment, which is now included in Supplementary Figure 2b (also attached below as Fig. I). This suggests that Pres^{TS} is closer to Prestin than to other SLC26 members. Note that the phylogenetic tree was generated based on an assumption that all residues can be spontaneously mutated (the default parameters in FastTree).

Supplementary Fig. I. A phylogenetic tree constructed for Prestin and other SLC26 family proteins: Pres^{TS}, human (Hs) SLC26A1, SLC26A2, SLC26A3, SLC26A4, SLC26A5, SLC26A6, SLC26A7, SLC26A8, SLC26A9, SLC26A10, SLC26A11, and zebrafish (Dr) SLC26A5, using FastTree with the default parameters.

Minor points:

l. 222 “upon the transition toward the outward-facing state (Cl⁻-bound hPres)”

Wording seems to suggest that the solved Cl⁻-bound hPres structure is outward-facing, while in fact it is occluded.

This is correct. Cl⁻-bound hPres adopts an occluded state, and it is uncertain whether hPres (and Pres^{TS}) could adopt an outward-facing state for its motor activity. To clarify this point, we changed the sentence to “upon the transition toward the Cl⁻-occluded state as captured in the recent prestin structures”

l. 203 onwards: for readability, better use a new subheading as this section deals with inter-domain interactions rather than salicylate effects.

Thank you for the comment. We added a new subheading, “Inter-domain interaction between the core and the gate domains” to the relevant section.

Specific comments by Reviewer #2:

Dear Authors,

Great work, it was a pleasure reading the revised version and the point-by-point discussion, which was performed very carefully.

Congratulations,

Dimitrios Fotiadis.

We appreciate Reviewer #2's favorable comments on our manuscript.

Reviewer #3 (Remarks to the Author):

I am happy with the clarifications, and changes done to the manuscript.

We appreciate Reviewer #3's favorable comments on our manuscript.

Reviewer #4 (Remarks to the Author):

The authors performed an additional 10 x 1 μs of MD simulations that provide sufficient evidence for

Cl⁻ binding to the putative binding site. They also performed additional analysis to more clearly define the role of the TM3/TM10 helix dipole moments to stabilize an anion in the binding pocket.

Other important changes:

A detailed discussion of the similarity between PresTS:Cl⁻ and PresTS:sulfate structures was added. [To this reviewer, the discussion in conjunction with the comparison to published structures suggests that the thermostabilization might have shifted (and locked) PresTS in an "down"-like state, although I would not dare to speculate that it resembles the physiological expanded state.]

Additional ITC measurements for the K_d of salicylate binding indicate that the binding site in hPres and PresTS bind salicylate at similar strength.

An interesting comparison between recently published prestin structures and the PresTS results was added.

We appreciate Reviewer #4's favorable comments on our manuscript.

I have a few remaining minor comments:

1) The One-sentence summary "Cryo-EM reveals the molecular mechanism of prestin." is still overselling the results and should be toned down.

We changed the one sentence summary to "Cryo-EM structures provide mechanistic insights into the prestin function"

2) The Introduction makes no mention of the 3 other prestin papers---from reading only about other SLC26 transporters one is lead to believe that there's nothing else known about prestin's structure. Please adhere to scholarly standards and cite the other papers and give an accurate picture of the current state of the field.

The last sentence in the introduction section was changed to "These structures, along with recently solved cryo-EM prestin structures, provide significant mechanistic insights toward the clarification of the voltage-driven motor function of prestin, which is responsible for the exquisite sensitivity and frequency selectivity of mammalian hearing."

3) L100 "These observations suggest that the consensus mutations collectively stabilized the structure of PresTS."

I don't follow this conclusion. How is it possible to infer from "loss of function" that Pres^{TS} was stabilized? Or maybe the paragraph-breaking is confusing and the concluding sentence was supposed to primarily refer to the SEC and cryo-EM structure determination data?

Note that the authors can make the case that their structure, even if non-functional, provides sufficient insights to generate new hypotheses. For example, they functionally probe residues in HgPres.

Thank you for the comment. This was the possibility we entertained. Since the validity of this speculation does not affect the outcomes of our study, we removed the last sentence “*These observations suggest that the consensus mutations collectively stabilized the structure of Pres^{TS}.*”.

4) L127 "probably corresponds to a Cl⁻ ion coordinated by R399 and S396"

With the somewhat large Cl⁻ - R399 distance and the finding of the importance of the helix dipoles, does it still make sense to call R399 a coordinating residue?

Thank you for the comment. As pointed out, Arg399 and the chloride ion are relatively apart in our structure, and the arginine sidechain makes only a weak interaction with the chloride ion by itself, but contributes to stabilization of the binding site, together with the surrounding residues. We changed the sentence to “probably corresponding to the Cl⁻ ion stabilized in this position by R399 and S396”.

5) On L154 "Notably, the coordination distance between the Cl⁻ ion and R399 found in our cryo-EM Pres^{TS} structure is 4.49 Å,..." a distance is quoted that does not match with the data in Suppl Fig 9, which presents distances > 6 Å for Cl⁻ - R399.

In Suppl Fig 9, the Cl⁻ - R399 distance is calculated to the C_{zeta} of the arginine. It is not clear if the 4.49 Å were calculated differently. Please clarify.

Presumably, it would be more meaningful to calculate the shortest distance for each simulation timestep between the anion and the positively charged guanidinium nitrogens because coordination distances are typically given for the ligand heavy atom. I suggest the authors add this distance $\min[d(\text{Cl}^-, N_{\eta 1}), d(\text{Cl}^-, N_{\eta 2})]$ to Suppl Fig 9.

We have revised Supplementary Figure 9, to depict time evolution of the nearest distance between the Cl⁻ ion and the N_η atoms of R399. The distance between the Cl⁻ ion and the closest N_η atom of R399 is distributed in 5-7 Å when the Cl⁻ ion is bound, showing that the Cl⁻ ion and Arg399 do not form a direct contact as described in the original manuscript. The distance is somewhat larger than that observed in the cryo-EM Pres^{TS} structure, 4.49 Å, presumably because of thermal fluctuation in the MD simulations at 300 K and the use of the native protein structure modeled from the Pres^{TS} one. We have revised the figure caption of Supplementary Fig. 9 as well.

6) typos

L161 "ans" -> "and"

L570 "perfoemd" -> "performed"

Thank you for the comment. We corrected these typos.

7) L210-212 "*Pres adopts an inward-open-like state, whereas hPres adopts an outward open-like conformation (Fig. 4a-c, d-left, Supplementary Fig.16b).*"

Just a comment: In the context of prestin I find it confusing to label the hPres:Cl⁻ structure as "outward-open" as this implies to me that the binding site has switched accessibility from the cytosolic to the extracellular solution. That makes sense for transporters but less so for prestin.

Thank you for the comment. As pointed out, the Cl⁻-bound hPres is an intermediate state, which is different from the outward-open state. We referred to the Cl⁻-bound hPres structure as "occluded-like intermediate conformation" in the revised manuscript.

8) L217 "*... structural determination of a Cl⁻-bound elongated state of prestin that was not captured in recent studies by others*"

There's no evidence presented that the PreTS:Cl⁻ structure is an "elongated state" (probably meant in the sense of "expanded"). There's no quantitative measure of similarity provided with putatively expanded structures or the direct change in area. Either provide evidence and clarify or remove.

This is an important comment. We removed our claim that the current structure represents an elongated state of prestin.

9) MD methods

Were really "constraints" used (i.e., the exact distance was held fixed with something like SHAKE) or were these "restraints" (e.g., harmonic pseudo bonds with a specific force constant)? Please clarify.

We used the restraints, not constraints. We have corrected the words in the method section (Molecular dynamics simulation) in the revised manuscript.

Were the electrostatics calculations performed with CPPTRAJ, AMBER, or something else, were periodic boundary conditions taken into account, and was the relative electric permittivity set to 1 (vacuum) or another value?

We calculated the decomposed contributions of the electrostatic interaction energy of the Cl⁻ ion by direct sums of the Coulombic interactions between the groups considered without periodic

images of the periodic boundary condition. The relative electric permittivity of the Coulombic interactions was set to 1. The electrostatic calculations were carried out with an in-house program.

It is noteworthy that the Ewald sum calculation with periodic images does not give well-defined decomposed electrostatic contributions. Unlike the total simulation system of which the electrostatic interaction energy was evaluated by the Ewald sum (PME) method, the total charge of a decomposed system (e.g., the Cl^- ion and the main chain part of an amino acid) is not necessarily neutral. In this case, first, a direct sum of the decomposed interaction energy of the non-neutral system in the periodic boundary condition diverges. Although the divergence is apparently removed in the Ewald sum method for the periodic boundary condition, a non-physical term of the background charge neutralizing the non-neutral system (i.e., the term of $k = 0$) is implicitly included in the decomposed energy. (For the charge neutral total system, the terms of the background charges cancel out.)

As the main purpose of the present analysis is comparison of the electrostatic interaction energies between different decomposed groups (e.g., helices and side-chain groups) of which the total charges are different, the use of a method based on the Ewald method which gives decomposed energies including the non-physical contributions is less appropriate. Given that the simulation box is sufficiently large (the lengths are more than 100 \AA) and the periodic images of the groups considered (i.e., the protein) are well separated, the direct sum calculation without the periodic images performed in the present analysis provides the most well-defined decomposed contributions of the electrostatic interaction energy.

The use of the relative electric permittivity of 1 in the calculation of the decomposed electrostatic contributions means that screening of the Coulombic interaction between the protein charges by polarization of the water environment and the protein is not considered. Unfortunately, such screening effects cannot be decomposed into the contributions of the protein groups required for the main purpose of the present analysis. We therefore employed the relative electric permittivity of 1 for simplicity.

We have clarified the method and the parameter used in the revised manuscript.

10) Fig 4c and Suppl Fig 16b

The schematic comparison of the different structures is very informative and a good example for clean graphical design. But it's not clear how the authors arrived at the ordering -- please describe in Methods or a caption what quantity was calculated to compare the different structures.

Thank you for the comment. The structures are superimposed at the gate domain, and the TM3/10 positions are illustrated based on the centrally located S398. This is clarified in the revised figure legends.